## Resource

# Mapping protein interactions of sodium channel Na_V1.7 using epitope-tagged gene-targeted mice

Alexandros H Kanellopoulos[1,†], Jennifer Koenig[1,†], Honglei Huang[2], Martina Pyrski[3] (ID),
Queensta Millet[1], Stéphane Lolignier[1,4], Toru Morohashi[1], Samuel J Gossage[1], Maude Jay[1],
John E Linley[1,5], Georgios Baskozos[6] (ID), Benedikt M Kessler[2], James J Cox[1], Annette C Dolphin[7],
Frank Zufall[3], John N Wood[1,*] & Jing Zhao[1,**] (ID)

## Abstract

The voltage-gated sodium channel Na_V1.7 plays a critical role in pain pathways. We generated an epitope-tagged Na_V1.7 mouse that showed normal pain behaviours to identify channel-interacting proteins. Analysis of Na_V1.7 complexes affinity-purified under native conditions by mass spectrometry revealed 267 proteins associated with Nav1.7 in vivo. The sodium channel β3 (Scn3b), rather than the β1 subunit, complexes with Nav1.7, and we demonstrate an interaction between collapsing-response mediator protein (Crmp2) and Nav1.7, through which the analgesic drug lacosamide regulates Nav1.7 current density. Novel Na_V1.7 protein interactors including membrane-trafficking protein synaptotagmin-2 (Syt2), L-type amino acid transporter 1 (Lat1) and transmembrane P24-trafficking protein 10 (Tmed10) together with Scn3b and Crmp2 were validated by co-immunoprecipitation (Co-IP) from sensory neuron extract. Nav1.7, known to regulate opioid receptor efficacy, interacts with the G protein-regulated inducer of neurite outgrowth (Gprin1), an opioid receptor-binding protein, demonstrating a physical and functional link between Nav1.7 and opioid signalling. Further information on physiological interactions provided with this normal epitope-tagged mouse should provide useful insights into the many functions now associated with the Na_V1.7 channel.

**Keywords** Na_V1.7; pain; protein–protein interactor; sensory neuron; sodium channel
**Subject Categories** Molecular Biology of Disease; Neuroscience
**The EMBO Journal (2018) 37: 427–445**

## Introduction

Pain is a major clinical problem. A 2012 National Health Interview Survey (NHIS) in the United States revealed that 25.3 million adults (11.2%) suffered from daily (chronic) pain and 23.4 million (10.3%) reported severe pain within a previous 3-month period (Nahin, 2015). Many types of chronic pain are difficult to treat as most available drugs have limited efficacy and can cause side effects. There is thus a huge unmet need for novel analgesics (Woodcock, 2009). Recent human and animal genetic studies have indicated that the voltage-gated sodium channel (VGSC) Na_V1.7 plays a crucial role in pain signalling (Nassar et al, 2004; Cox et al, 2006; Dib-Hajj et al, 2013), highlighting Na_V1.7 as a promising drug target for development of novel analgesics (Habib et al, 2015; Emery et al, 2016).

VGSCs consist of a large pore-forming α-subunit (~260 kDa) together with associated auxiliary β-subunits (33–36 kDa) and play a fundamental role in the initiation and propagation of action potentials in electrically excitable cells. Nine isoforms of α-subunits (Na_V1.1–1.9) that display distinct expression patterns, and variable channel properties have been identified in mammals (Frank & Catterall, 2003). Na_V1.7, encoded by the gene SCN9A in humans, is selectively expressed peripherally in dorsal root ganglion (DRG), trigeminal ganglia and sympathetic neurons (Toledo-Aral et al, 1997; Black et al, 2012), as well as in the central nervous system (Weiss et al, 2011; Branco et al, 2016). As a large membrane ion channel, Na_V1.7 produces a fast-activating, fast-inactivating and slowly repriming current (Klugbauer et al, 1995), acting as a threshold channel to contribute to the generation and propagation of action potentials by amplifying small sub-threshold depolarisations (Rush et al, 2007). The particular electrophysiological characteristics of Na_V1.7 suggest that it plays a key role in initiating action

1   Molecular Nociception Group, WIBR, University College London, London, UK
2   TDI Mass Spectrometry Laboratory, Target Discovery Institute, University of Oxford, Oxford, UK
3   Center for Integrative Physiology and Molecular Medicine, Saarland University, Homburg, Germany
4   Université Clermont Auvergne, Inserm U1107 Neuro-Dol, Pharmacologie Fondamentale et Clinique de la Douleur, Clermont-Ferrand, France
5   Neuroscience, IMED Biotech Unit, AstraZeneca, Cambridge, UK
6   Division of Bioscience, University College London, London, UK
7   Department of Neuroscience, Physiology and Pharmacology, University College London, London, UK
    *Corresponding author. Tel: +44 207 6796 954; E-mail: j.wood@ucl.ac.uk
    **Corresponding author. Tel: +44 207 6790 959; E-mail: jing02.zhao@ucl.ac.uk
    † These authors contributed equally to this work

potentials in response to depolarisation of sensory neurons by noxious stimuli (Habib *et al*, 2015). In animal studies, our previous results demonstrated that conditional Na$_V$1.7 knockout mice have major deficits in acute, inflammatory and neuropathic pain (Nassar *et al*, 2004; Minett *et al*, 2012). Human genetic studies show that loss-of-function mutations in Na$_V$1.7 lead to congenital insensitivity to pain, whereas gain-of-function mutations cause a range of painful inherited disorders (Cox *et al*, 2006; Dib-Hajj *et al*, 2013). Recent studies also show that Na$_V$1.7 is involved in neurotransmitter release in both the olfactory bulb and spinal cord (Weiss *et al*, 2011; Minett *et al*, 2012). Patients with recessive Na$_V$1.7 mutations are normal (apart from being pain-free and anosmic), suggesting that selective Na$_V$1.7-blocking drugs are therefore likely to be effective analgesics with limited side effects (Cox *et al*, 2006).

Over the past decade, an enormous effort has been made to develop selective Na$_V$1.7 blockers. Efficient and selective Na$_V$1.7 antagonists have been developed; however, Na$_V$1.7 antagonists require co-administration of opioids to be fully analgesic (Minett *et al*, 2015). Na$_V$1.7 thus remains an important analgesic drug target and an alternative strategy to target Na$_V$1.7 could be to interfere with either intracellular trafficking of the channel to the plasma membrane, or the downstream effects of the channel on opioid peptide expression. Although some molecules, such as Crmp2, ubiquitin ligase Nedd4-2, fibroblast growth factor 13 (FGF13) and PDZ domain-containing protein Pdzd2, have been reported to associate with the regulation of trafficking and degradation of Na$_V$1.7, the entire protein–protein interaction network of Na$_V$1.7 still needs to be defined (Shao *et al*, 2009; Dustrude *et al*, 2013; Laedermann *et al*, 2013; Bao, 2015; Yang *et al*, 2017). Affinity purification (AP) and targeted tandem affinity purification (TAP) combined with mass spectrometry (MS) is a useful method for mapping the organisation of multiprotein complexes (Angrand *et al*, 2006; Wildburger *et al*, 2015). Using the AP-MS or TAP-MS method to characterise protein complexes from transgenic mice allows the identification of complexes in their native physiological environment in contact with proteins that might only be specifically expressed in certain tissues (Fernandez *et al*, 2009). The principal aim of this study was to identify new protein interaction partners and networks involved in the trafficking and regulation of the sodium channel Na$_V$1.7 using mass spectrometry and our recently generated epitope (TAP)-tagged Na$_V$1.7 knock-in mouse. Such information should throw new light on the mechanisms through which Na$_V$1.7 regulates a variety of physiological systems, as well as propagating action potentials, especially in pain signalling pathways.

# Results

### The TAP-tag does not affect Na$_V$1.7 channel function

Prior to generating the TAP-tagged Na$_V$1.7 knock-in mouse, we tested the channel function of TAP-tagged Na$_V$1.7 in HEK293 cells by establishing a HEK293 cell line stably expressing TAP-tagged Na$_V$1.7. A 5-kDa TAP-tag consisting of a poly-histidine affinity tag (HAT) and a 3x FLAG-tag in tandem (Terpe, 2003), separated by a unique TEV protease cleavage site, was fused to the C-terminus of Na$_V$1.7 (Fig 1A). The expression of TAP-tagged Na$_V$1.7 was detected with immunocytochemistry using an anti-FLAG antibody. The result showed that all the HEK293 cells expressed TAP-tagged Na$_V$1.7

(Fig 1B). The channel function of TAP-tagged Na$_V$1.7 was examined with electrophysiological analysis, and the result showed that all the cells presented normal functional Na$_V$1.7 currents (Fig 1C). Activation and fast inactivation data were identical for wild-type Na$_V$1.7 and TAP-tagged Na$_V$1.7 (Fig 1D) demonstrating that the TAP-tag does not affect the channel function of Na$_V$1.7.

### Generation of a TAP-tagged Na$_V$1.7 mouse

We used a conventional gene targeting approach to generate an epitope-tagged Na$_V$1.7 mouse. The gene targeting vector was constructed using an *Escherichia coli* recombineering-based method (Bence *et al*, 2005; Fig EV1). A TAP-tag, which contains a HAT domain, a TEV cleavage site and 3x FLAG-tags (Fig 1A), was inserted into the open reading frame at the 3′-end prior to the stop codon in exon 27 of Na$_V$1.7 (NCBI Reference: NM_001290675; Fig 2A). The final targeting vector construct containing a 5′ homology arm (3.4 kb), a TAP-tag, neomycin cassette and a 3′ homology arm (5.8 kb; Fig 2B) was transfected into the 129/Sv embryonic stem (ES) cells. Twelve colonies with the expected integration (targeting efficiency was 3.5%) were detected by Southern blot analyses (Fig 2C). Germline transmission and intact TAP-tag insertion after removal of the neomycin (neo) cassette were confirmed with Southern blot, PCR and RT–PCR, respectively (Fig 2D–F). This mouse line is henceforth referred to as Na$_V$1.7$^{TAP}$.

### Na$_V$1.7$^{TAP}$ mice have normal pain behaviour

The homozygous Na$_V$1.7$^{TAP}$ knock-in mice (KI) were healthy, fertile and apparently normal. Motor function of the mice was examined with the Rotarod test. The average time that KI animals stayed on the rod was similar to the WT mice (Fig 3A), suggesting there is no deficit on motor function in the KI mice. The animal responses to low-threshold mechanical, acute noxious thermal, noxious mechanical stimuli and acute peripheral inflammation were examined with von Frey filaments, Hargreaves' test, Randall–Selitto apparatus and formalin test, respectively. The results showed that TAP-tagged KI mice had identical responses to these stimuli compared to the littermate WT control mice (Fig 3B–E), indicating Na$_V$1.7$^{TAP}$ mice have normal acute pain behaviour.

### Expression pattern of TAP-tagged Na$_V$1.7 in the nervous system

We used immunohistochemistry to examine the expression pattern of TAP-tagged Na$_V$1.7 in the nervous system of Na$_V$1.7$^{TAP}$ mice. The results showed that the FLAG-tag was expressed in the olfactory bulb, with strong staining visible in the olfactory nerve layer, the glomerular layer, and in the accessory olfactory bulbs (Fig 4A–D), consistent with previous results (Weiss *et al*, 2011). In the brain, FLAG-tag expression was present in the medial habenula, the anterodorsal thalamic nucleus, the laterodorsal thalamic nucleus and in the subfornical organ that is located in the roof of the dorsal third ventricle (Fig 4E and F) and is involved in the control of thirst (Oka *et al*, 2015). The FLAG-tag was also present in neurons of the posterodorsal aspect of the medial amygdala and in the hypothalamus in neurons of the arcuate nucleus (Fig 4G–I) as confirmed in a recent study (Branco *et al*, 2016). A clear staining appeared in

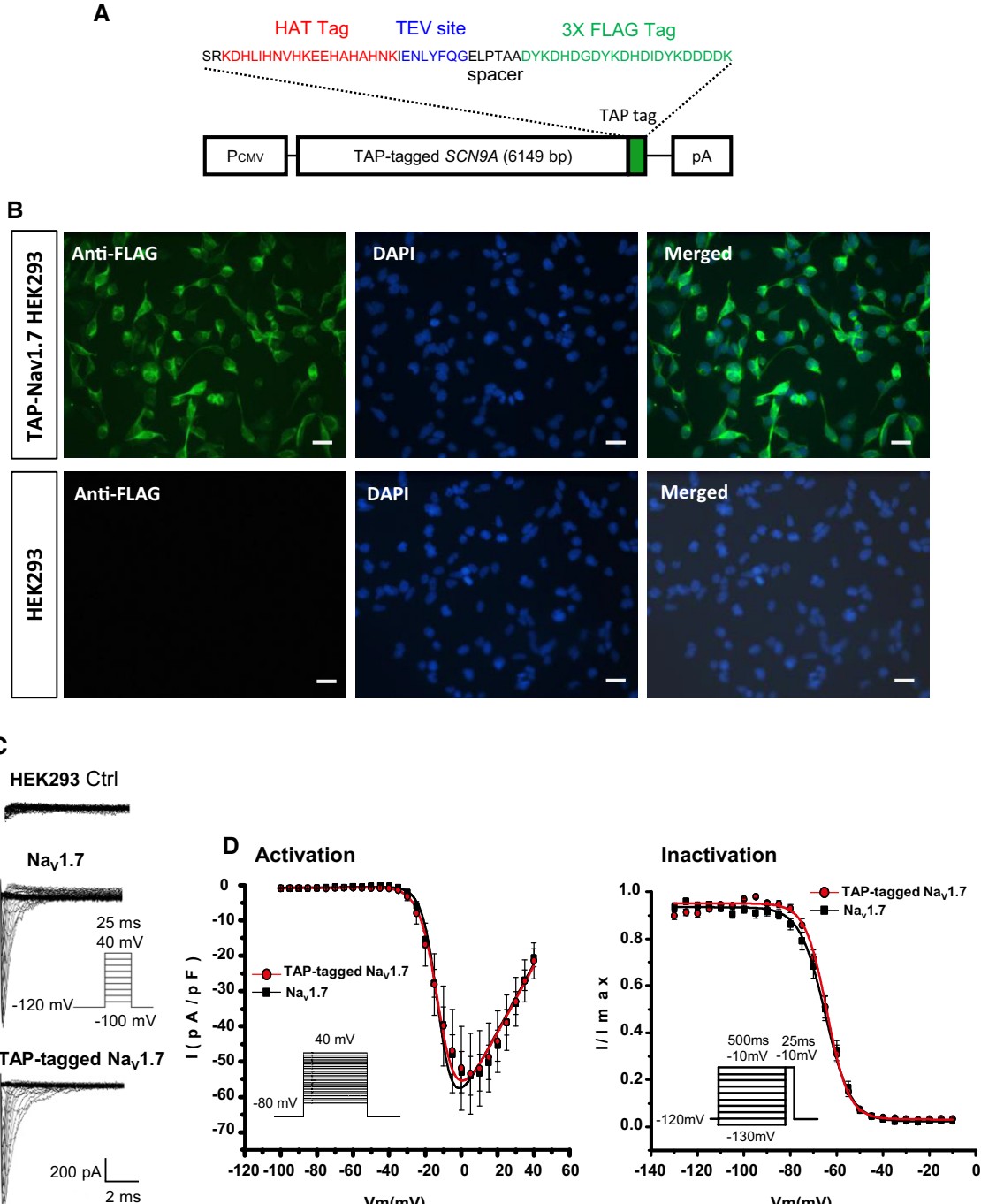

**Figure 1.  Characterisation of TAP-tagged Na$_V$1.7 in HEK293 cells.**

A    The diagram of TAP-tagged *SCN9A* cDNA construct used to establish the stably expressing TAP-tagged Na$_V$1.7 HEK293 cell line. A sequence encoding a TAP-tag was cloned immediately upstream of the stop codon of *SCN9A* coding the wild-type Na$_V$1.7.

B    Left panel: Representative immunohistochemistry with an anti-FLAG antibody (in green) on HEK293 cells stably expressing TAP-tagged Na$_V$1.7 (top panel) and parental HEK293 cells (bottom panel). Middle panel: Cell nuclei were costained with DAPI (blue). Right panel: the left panels were merged to the middle panels. Scale bar = 25 μm.

C    Representative current traces recorded from HEK293 cells stably expressing either the wild-type Na$_V$1.7 or the TAP-tagged Na$_V$1.7 in response to depolarisation steps from −100 to 40 mV.

D    Left panel: I(V) curves obtained in Na$_V$1.7- and TAP-tagged Na$_V$1.7-expressing cells using the same protocol as in (C), showing no significant difference in voltage of half-maximal activation (V$_{1/2}$; −9.54 ± 1.08 mV for Na$_V$1.7, n = 10; and −8.12 ± 1.07 mV for TAP-tagged Na$_V$1.7, n = 14; P = 0.3765, Student's t-test) and reversal potential (V$_{rev}$; 61.73 ± 3.35 mV for Na$_V$1.7, n = 10; and −8.12 ± 1.07 mV for TAP-tagged Na$_V$1.7, n = 14; P = 0.9114, Student's t-test). Right panel: Voltage dependence of fast inactivation, assessed by submitting the cells to a 500-ms prepulse from −130 to −10 mV prior to depolarisation at −10 mV. No significant difference in the voltage of half inactivation was observed between the two cell lines (V$_{1/2}$ of −64.58 ± 1.32 mV for Na$_V$1.7, n = 10; and −64.39 ± 1.06 for TAP-tagged Na$_V$1.7, n = 14; P = 0.9100, Student's t-test). All data are mean ± SEM.

    

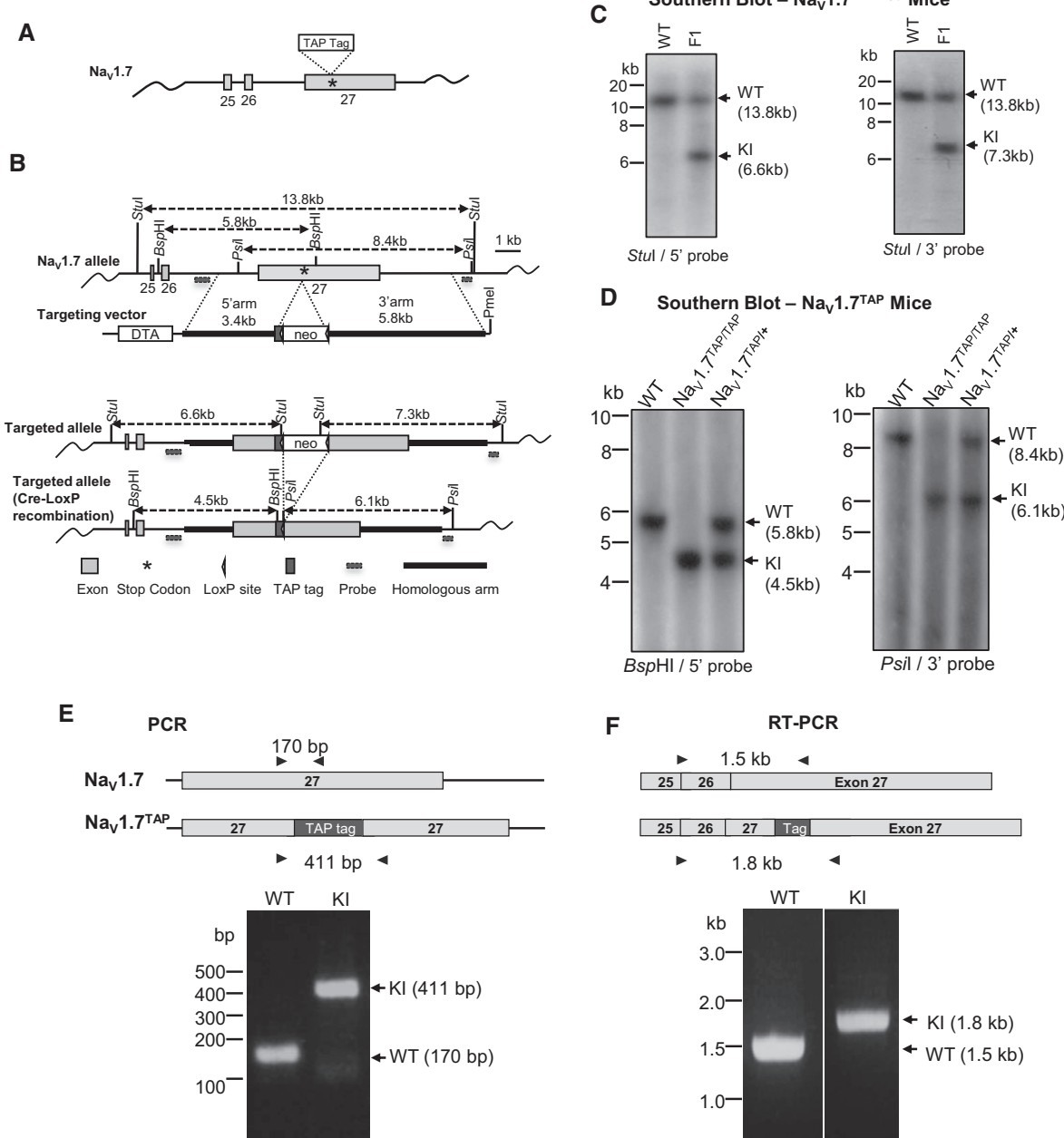

**Figure 2.  Generation of TAP-tagged Na$_V$1.7 knock-in mice.**

A  The location of TAP-tag in the Na$_V$1.7 locus. A sequence encoding a TAP-tag peptide comprised of a HAT domain, TEV cleavage site and 3 FLAG-tags was inserted immediately prior to the stop codon (indicated as a star) at the extreme C-terminus of Na$_V$1.7.

B  Schematic diagrams of the targeting strategy. Grey boxes represent Na$_V$1.7 exons (exon numbers are indicated on the box), black box represents TAP-tag, thick black lines represent homologous arms, and the small triangle box represents the single LoxP site, respectively. The positions of the external probes used for Southern blotting are indicated in the diagram. Neomycin (neo), DTA expression cassettes and restriction enzyme sites and expected fragment sizes for Southern blotting are also indicated.

C  Southern blot analysis of genomic DNA from Founder 1 mice (TAP-tagged Na$_V$1.7 carrying the neo cassette). Genomic DNA was digested with *Stu*I and was then hybridised with either 5′ or 3′ external probe. Wild-type (WT) allele was detected as a 13.8 kb fragment using either 5′ or 3′ probes. Knock-in allele (KI) was detected as either a 6.6-kb (5′ probe) or a 7.3-kb (3′ probe) fragment.

D  Southern blot analysis of TAP-tagged Na$_V$1.7 allele after Cre recombination. Genomic DNA was digested either with *Bsp*HI or *Psi*I and was then hybridised with either 5′ (*Bsp*HI) or 3′ (*Psi*I) external probe. WT alleles were detected as 5.8 kb (5′ probe) and 8.4 kb (3′ probe) fragments, respectively. The neo-deleted TAP-tagged Na$_V$1.7 alleles were detected as 4.5-kb (5′ probe) and 6.1-kb (3′ probe) fragments, respectively.

E  Genotyping analysis by PCR. Representative result of the PCR screening of Na$_V$1.7$^{TAP}$ mice showing the 411-bp band (KI allele) and the 170-bp band (WT allele). The location of primers used for PCR is indicated with black arrows.

F  TAP-tagged Na$_V$1.7 expression analysis with RT–PCR. Total RNA was isolated from DRG of Na$_V$1.7$^{TAP}$ mice, and cDNA synthesis was primed using oligo-dT. PCR was performed with the primers indicated with black arrows. A 1.5-kb WT band and a 1.8-kb KI band were detected from either littermate WT control animals or Na$_V$1.7$^{TAP}$ KI mice, respectively.

    

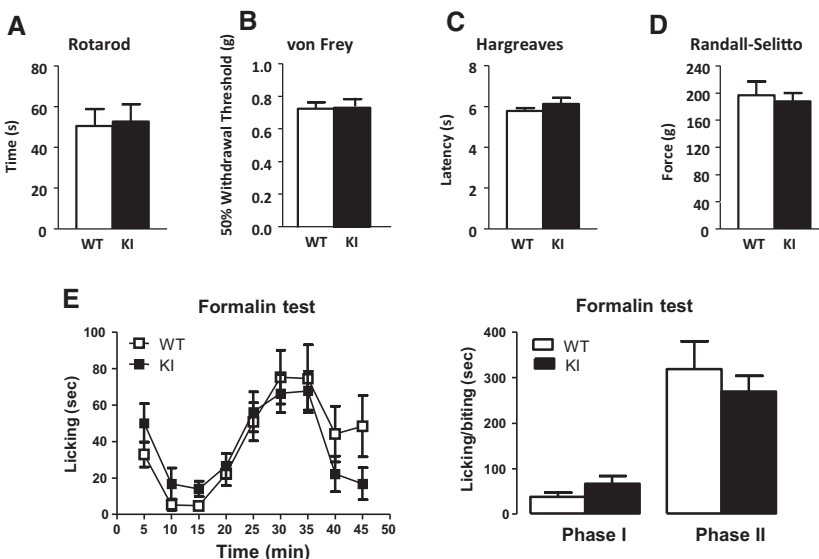

**Figure 3. Pain behaviour tests.**

A Rotarod test showed no motor deficits in TAP-tagged Na$_V$1.7 animals ($n = 7$, WT; $n = 7$, KI; $P = 0.8619$, Student's $t$-test).

B Responses to low-threshold mechanical stimulation by von Frey filaments were normal in the KI mice ($P = 0.9237$, Student's $t$-test).

C Hargreaves' apparatus demonstrated identical latencies of response to thermal stimulation ($n = 7$, WT; $n = 7$, KI; $P = 0.3038$, Student's $t$-test).

D Acute mechanical pressure applied with the Randall–Selitto apparatus demonstrated identical behaviour in KI and WT mice ($n = 7$, WT; $n = 7$, KI; $P = 0.7124$, Student's $t$-test).

E Formalin test. Licking/biting response to acute peripheral inflammation induced by intraplantar injection of 5% formalin in hind-paw was recorded. Left panel, the time course of development of the response of KI mice (black squares) and WT littermate controls (white squares) showed similar response patterns ($n = 10$, WT; $n = 7$, KI; $P = 0.2226$, two-way ANOVA). Right panel, the early (0–10 min) and late (10–45 min) phases of the formalin response in KI and WT mice showed similar responses ($P = 0.1690$, first phase; $P = 0.5017$, second phase; Student's $t$-test) between KI and WT mice.

Data information: All data are mean ± SEM.

neurons of the substantia nigra reticular region and the red nucleus magnocellular region of the midbrain, and in neurons of the pontine nuclei located in the hindbrain (Fig 4J–L). In the spinal cord, FLAG-tag expression was visible in the superficial layer of the dorsal horn. Costaining of the spinal cord with isolectin B4 (IB4), a marker for the inner part of lamina II, showed that the TAP-tagged Na$_V$1.7 was expressed in laminae I, II and III (Fig 4M–O). In the PNS, however, there was no positive FLAG-tag staining found in DRG, sciatic nerve or skin nerve terminals (data not shown). This could be because of masking of the tag in the PNS, preventing the antibody binding. We also examined the TAP-tagged Na$_V$1.7 expression pattern in

different tissues with Western blot using an anti-HAT antibody. TAP-tagged Na$_V$1.7 bands were present in olfactory bulb, hypothalamus, spinal cord, sciatic nerve and DRG, but not detectable in cortex, cerebellum, skin, lung, heart and pancreas (Fig 5E).

## Optimisation of single-step- and tandem affinity purification of TAP-tagged Na$_V$1.7

The TAP-tag on Na$_V$1.7 offered the possibility of two consecutive affinity purifications, a single-step affinity purification (ss-AP; steps 1 and 2 in Fig 5A) and a tandem affinity purification (steps 1–4 in

**Figure 4. Immunohistochemistry of FLAG-tag expression in the central nervous system (CNS).**

A–D In the main olfactory bulb (MOB) (A), FLAG-tag expression (in green) is visible in the olfactory nerve layer (ONL) and in the glomerular layer (GL) in TAP-tagged Na$_V$1.7 knock-in mice (KI) but not in the littermate wild-type controls (WT). The white square box in (A) is shown with high magnification in (B). In the posterior olfactory bulb, staining is also evident in the (C) accessory olfactory bulb (AOB). Staining is absent in the (D) MOB and AOB of wild-type control mice.

E, F FLAG-tag expression is present (E) in the medial habenula (MHb, arrow), the anterodorsal thalamic nucleus (AD, arrow), the laterodorsal thalamic nucleus (LD, dotted line) and (F) in the subfornical organ (SFO, arrow) located in the roof of the dorsal third ventricle.

G–I FLAG-tag expression is present (G) in neurons of (H) the posterodorsal aspect of the medial amygdala (MePD, arrow, dotted line) and in (I) the hypothalamus in neurons of the arcuate nucleus (Arc, arrow, dotted line).

J–L FLAG-tag expression is present (J) in neurons of the substantia nigra reticular part (SNR) and (K) the red nucleus magnocellular part (RMC) of the midbrain, and (L) in neurons of the pontine nuclei (Pn) located in the hindbrain.

M, N The cross section of lumbar spinal cord (L4) is labelled with anti-FLAG (in red). FLAG-tag expression is present in laminae I, II and III in spinal cord of KI mice (N) but not in spinal cord of WT mice (M).

O The cross sections of spinal cord of KI mice were costained with laminae II marker IB4 (in green).

Data information: Sketches on the left illustrate the CNS regions and bregma levels (in mm) of the fluorescence images shown on the right. Scale bars: 500 μm (A, C, D, G); 250 μm (E, J, K, M–O); 100 μm (B, F, I, L); 50 μm (H). cp, cerebral peduncle; CTX, cortex; DM, dorsomedial hypothalamic nucleus; EPL, external plexiform layer; ME, median eminence; opt, optic tract; sm, stria medullaris; TH, thalamus; VMH, ventromedial hypothalamic nucleus; 3V, third ventricle.

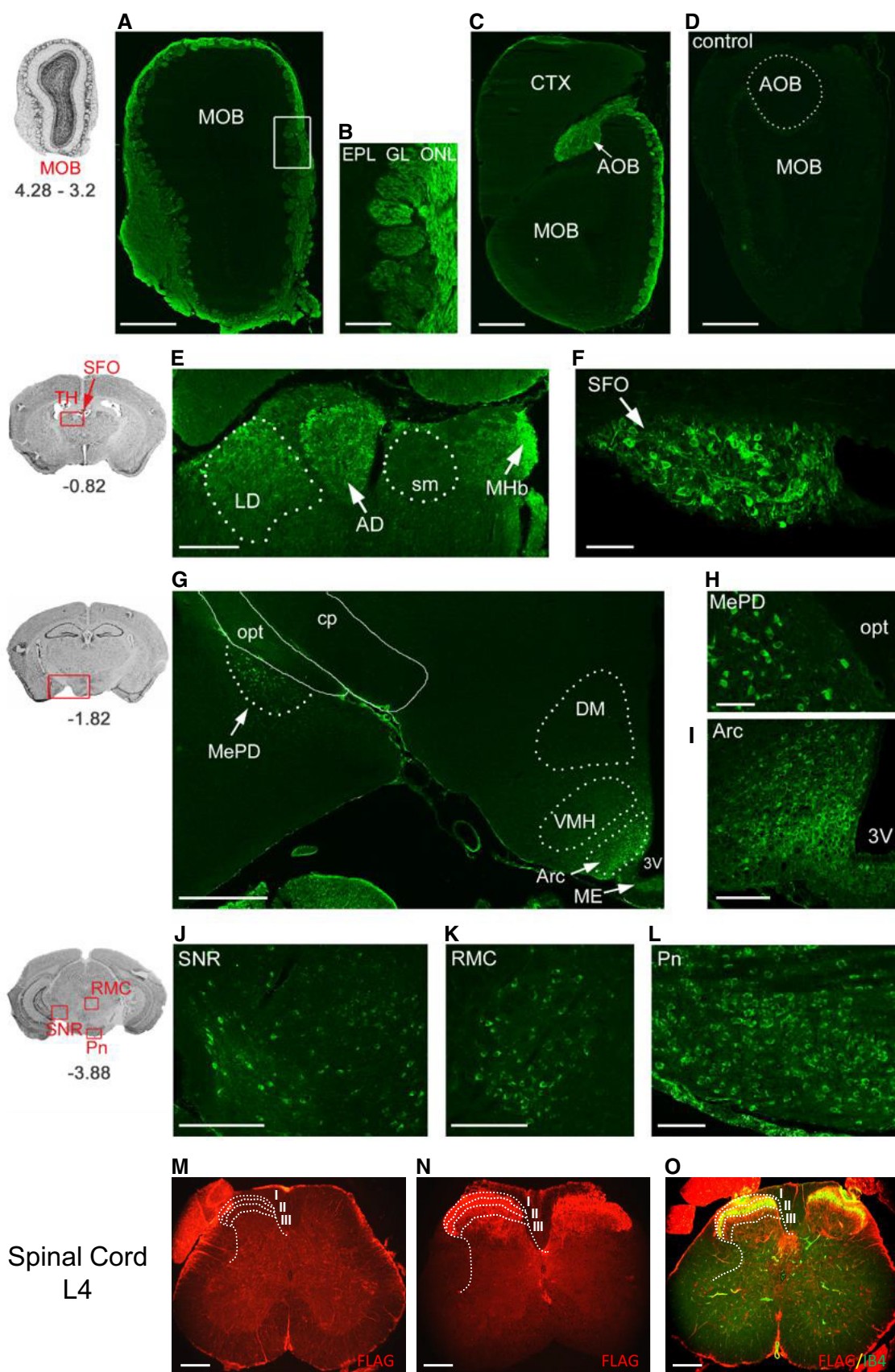

**Figure 4.**

**Figure 5.  Optimisation of single-step and tandem affinity purification, validation of identified protein–protein interactors and tissue expression pattern of TAP-tagged Na$_V$1.7 in Na$_V$1.7$^{TAP}$ mice.**

A  Schematic illustrating the affinity purification (ss-AP and TAP) procedure using the tandem affinity tags separated with a TEV cleavage site.

B  The proteins from DRG and olfactory bulbs were extracted in 1% CHAPS lysis buffer. After single-step and tandem affinity purification, TAP-tagged Na$_V$1.7 was detected using Western blotting with anti-HAT antibody.

C  The proteins from different tissues including hypothalamus, sciatic nerve, spinal cord, olfactory bulbs and DRG from KI mice, and pooled tissues from WT mice were extracted in 1% CHAPS lysis buffer. After single-step affinity purification, TAP-tagged Na$_V$1.7 was detected using Western blotting with anti-HAT antibody.

D  The interaction between TAP-tagged Na$_V$1.7 and identified Na$_V$1.7 protein–protein interactors including Scn3b, Syt2, Crmp2, Gprin1, Lat1 and Tmed10 was validated using a co-immunoprecipation *in vitro* system. The expression vectors containing cDNA of validated genes were cloned and transfected into a HEK293 cell line stably expressing TAP-tagged Na$_V$1.7. After transfection, TAP-tagged Na$_V$1.7 complexes were immunoprecipitated with anti-FLAG antibody, and the selected candidates were detected with their specific antibody using Western blotting. The results showed the expected sizes of Scn3b (32 kDa), Syt2 (44 kDa), Crmp2 (70 kDa), Gprin1 (two isoforms: 80 kDa and 110 kDa), Lat1 (57 kDa) and Tmed10 (21 kDa).

E  Tissue expression pattern of TAP-tagged Na$_V$1.7. The proteins were extracted from different tissues in both KI and WT littermate control mice and anti-FLAG used to detect TAP-tagged Na$_V$1.7 using Western blotting. Anti β-tubulin was used as a loading control.

F  The validation of selected Na$_V$1.7 protein interactor candidates with Nav1.7 endogenous expressing DRG tissue. First, the proteins from DRG of TAP-tagged Na$_V$1.7 mice were extracted in 1% CHAPS lysis buffer. Na$_V$1.7 complexes were then immunoprecipitated by anti-FLAG M2 magnetic beads. Thirteen Na$_V$1.7 interactor candidates including Scn3b (32 kDa), Syt2 (44 kDa), Crmp2 (70 kDa), Gprin1 (110 kDa), Lat1 (57 kDa), Tmed10 (21 kDa), Akap12 (191 kDa), Nfasc (138 kDa), Ntm (38 kDa), Kif5b (110 kDa), Ank3 (243 kDa) and Pebp1 (23 kDa) were detected with their specific antibodies using Western blotting.

G  Co-immunoprecipitation of Na$_V$1.7 with Ca$_V$2.2. Left panel shows negative Western blot results for pull-down of transiently transfected HA-tagged Ca$_V$2.2 from TAP-tagged Na$_V$1.7 complex (HAT antibody for detection) in TAP-tagged Na$_V$1.7 HEK293 stable cell line. Right panel shows control blot from whole-cell lysate of HA-tagged Ca$_V$2.2 and TAP-tagged Na$_V$1.7.

Source data are available online for this figure.

Fig 5A). CHAPS and DOC lysis buffers were evaluated to solubilise TAP-tagged Na$_V$1.7 and its protein complex from tissues, for example DRG, spinal cord, olfactory bulbs and hypothalamus, in the co-IP system. CHAPS is a non-denaturing zwitterionic detergent, commonly used to extract membrane proteins in their native conformation. In comparison with strong anionic detergents like SDS, CHAPS preserves protein–protein interactions and is compatible with downstream applications such as mass spectrometry. The result showed that TAP-tagged Na$_V$1.7 from DRG and olfactory bulbs was clearly solubilised and precipitated by both purifications —ss-AP and tandem affinity purification in 1% CHAPS buffer (Fig 5B). Also, the result from ss-AP showed that TAP-tagged Na$_V$1.7 could be immunoprecipitated from hypothalamus, sciatic nerve, spinal cord, olfactory bulb and DRG of KI mice, but not from these tissues of WT control mice (Fig 5C). However, the DOC lysis buffer, which was used to investigate the TAP-tagged PSD-95 protein complex (Fernandez *et al*, 2009), did not solubilise TAP-tagged Na$_V$1.7 from mouse tissue (data not shown).

## Identification of TAP-tagged Na$_V$1.7-associated complexes by AP-MS

We next identified the components of Na$_V$1.7 complexes using ss-AP followed by Liquid chromatography–tandem mass spectrometry (LC-MS/MS). Briefly, the TAP-tagged Na$_V$1.7 complexes were extracted from DRG, spinal cord, olfactory bulb and hypothalamus using ss-AP (see Materials and Methods). In total, 189,606 acquired spectra from 12 samples, in which each group (KI and WT) contains six biological replicate samples and each sample was from one mouse, were used for protein identification; 1,252 proteins were identified with a calculated 0.96% false discovery rate (FDR); 267 proteins (Table EV1) met those criteria and were shortlisted based on the criteria described in Materials and Methods. The proteins only appearing in Na$_V$1.7$^{TAP}$ mice and the representatively selected proteins are listed in Table 1. PANTHER cellular component analysis of these 267 proteins revealed eight different cellular components (Fig 6A). These proteins were further classified into 22 groups based on their function, including 12 membrane-trafficking proteins, 23 enzyme modulators and four transcription factors (Fig 6B).

## Validation of TAP-tagged Na$_V$1.7-interacting proteins using co-IP

The physical interactions between Na$_V$1.7 and interacting protein candidates were assessed using co-IP with DRG tissue extracts from TAP-tagged Na$_V$1.7 mice. A number of candidates of interest, such as Scn3b, Syt2, Lat1, Tmed10, Gprin1, Crmp2, isoform 2 of A-kinase anchor protein 12 (Akap12), neurofascin (Nfasc), neurotrimin (Ntm), kinesin-1 heavy chain (Kif5b), ankyrin-3 (Ank3) and phosphatidylethanolamine-binding protein 1 (Pebp1), were chosen from the Na$_V$1.7-associated protein list previously identified by MS (Table 1 and Table EV1). After ss-AP, the Na$_V$1.7 complexes were separated by SDS–PAGE and the protein interactors of Na$_V$1.7 were detected by Western blotting. Our results show that all 12 candidates were detected by their specific antibodies (Fig 5F).

We also validated six candidates (Scn3b, Syt2, Crmp2, Gprin1, Lat1 and Tmed10) using co-IP in an *in vitro* system by co-expressing the candidates in a TAP-tagged Na$_V$1.7 stable cell line. The mammalian expression vectors carrying cDNAs of these candidates were transfected into a TAP-tagged Na$_V$1.7 stable HEK293 cell line; 48 h of post-transfection, the proteins in the transfected cells were extracted. The TAP-tagged Na$_V$1.7 multiprotein complexes were then immunoprecipitated with anti-FLAG antibody and analysed with Western blot using different specific antibodies against those selected candidates. The Western blot showed that all six candidates were detected with the expected sizes on the blot (Fig 5D), confirming that these candidates are contained in the Na$_V$1.7 complex. In summary, all these potential interacting proteins selected for further validation were confirmed as interactors by co-IP, giving confidence in the Nav1.7$^{TAP}$ mass spectrometry list.

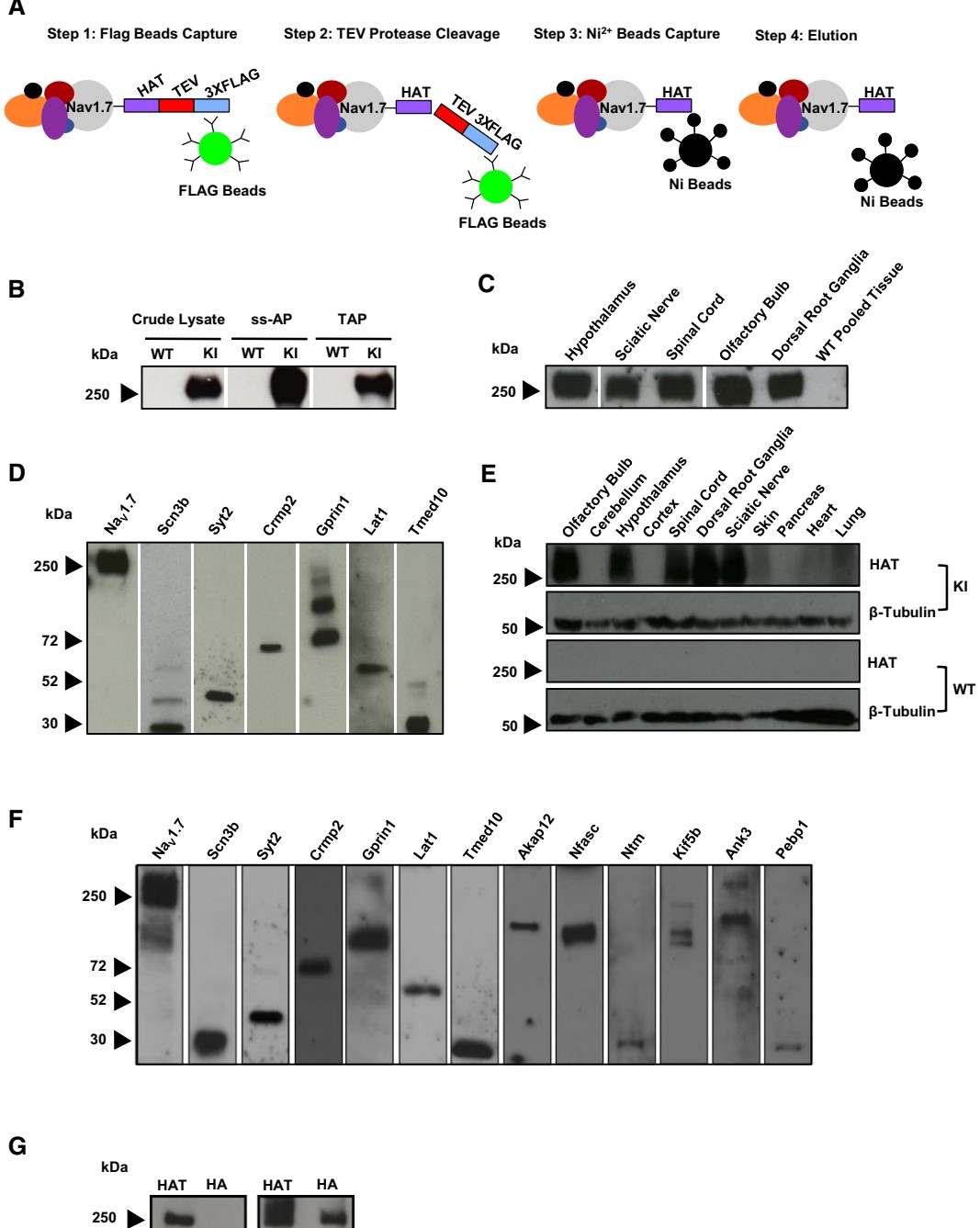

**Figure 5.**

### Functional characterisation of Crmp2

We confirmed that Tap-tagged Nav1.7 binds directly to Crmp2, the presumed target of the anti-epileptic and analgesic drug laco-samide, based on binding studies (Wilson & Khanna, 2015). We sought to evaluate the possible electrophysiological consequences of Crmp2 binding to Na$_V$1.7 and in addition to understand the relevance of Crmp2 to the action of lacosamide. Transfection of Crmp2 into a Na$_V$1.7 stable HEK293 cell line revealed a nearly twofold increase in sodium current density (Fig 7A–C), consistent

with Crmp2 acting directly as a transporter for Na$_V$1.7 (Dustrude et al, 2013). Next, we sought to investigate the effect of lacosamide on sodium currents. In cells not transfected with Crmp2, Na$_V$1.7 currents displayed very small changes in current density under our recording conditions following prolonged 5-h exposure to lacosamide (Fig 7A and B). Interestingly, following 5-h incubation with lacosamide in Crmp2-transfected cells, a complete rever-sal in the increase in Na$_V$1.7 current density provoked by Crmp2 was observed (Fig 7A and D). This shows an interaction between Na$_V$1.7 and Crmp2 and demonstrates that actions of lacosamide

**Table 1.** Identified Na$_V$1.7-associated proteins (only appearing in KI group + selected candidates).

| Gene symbol | Protein name | UniProt Acc | Normalised spectral index quantity | | |
|---|---|---|---|---|---|
| | | | KI (Mean) | WT (Mean) | Ratio (KI/WT) |
| *Proteins appearing in KI group only* | | | | | |
| *Scn3b* | Sodium channel subunit beta-3 | Q8BHK2 | 2.64E−06 | 0.00E+00 | KI only |
| *Bola2* | BolA-like protein 2 | Q8BGS2 | 2.28E−06 | 0.00E+00 | KI only |
| *Psma7* | Proteasome subunit alpha type-7 | Q9Z2U0 | 1.77E−06 | 0.00E+00 | KI only |
| *Psma1* | Proteasome subunit alpha type-1 | Q9R1P4 | 1.70E−06 | 0.00E+00 | KI only |
| *Homer2* | Isoform 2 of Homer protein homolog 2 | Q9QWW1-2 | 1.22E−06 | 0.00E+00 | KI only |
| *Fga* | Isoform 2 of Fibrinogen alpha chain | E9PV24-2 | 1.06E−06 | 0.00E+00 | KI only |
| *Ndufv3* | NADH dehydrogenase [ubiquinone] flavoprotein 3, mitochondrial | Q8BK30 | 9.48E−07 | 0.00E+00 | KI only |
| *Psma5* | Proteasome subunit alpha type-5 | Q9Z2U1 | 9.22E−07 | 0.00E+00 | KI only |
| *Erh* | Enhancer of rudimentary homolog | P84089 | 9.12E−07 | 0.00E+00 | KI only |
| *Psmb2* | Proteasome subunit beta type-2 | Q9R1P3 | 8.92E−07 | 0.00E+00 | KI only |
| *Rp2* | Isoform 2 of Protein XRP2 | Q9EPK2-2 | 8.67E−07 | 0.00E+00 | KI only |
| *Pebp1* | Phosphatidylethanolamine-binding protein 1 | P70296 | 8.32E−07 | 0.00E+00 | KI only |
| *Plekhb1* | Isoform 2 of Pleckstrin homology domain-containing family B member 1 | Q9QYE9-2 | 8.18E−07 | 0.00E+00 | KI only |
| *Omp* | Olfactory marker protein | Q64288 | 6.62E−07 | 0.00E+00 | KI only |
| *Ndufb10* | NADH dehydrogenase [ubiquinone] 1 beta subcomplex subunit 10 | Q9DCS9 | 6.24E−07 | 0.00E+00 | KI only |
| *Fabp7* | Fatty acid-binding protein, brain | P51880 | 5.87E−07 | 0.00E+00 | KI only |
| *Apoo* | Apolipoprotein O | Q9DCZ4 | 5.53E−07 | 0.00E+00 | KI only |
| *Psmb5* | Proteasome subunit beta type-5 | O55234 | 4.79E−07 | 0.00E+00 | KI only |
| *Psmd12* | 26S proteasome non-ATPase regulatory subunit 12 | Q9D8W5 | 4.47E−07 | 0.00E+00 | KI only |
| *Mlst8* | Target of rapamycin complex subunit LST8 | Q9DCJ1 | 4.22E−07 | 0.00E+00 | KI only |
| *Flna* | Filamin-A | Q8BTM8 | 4.00E−07 | 0.00E+00 | KI only |
| *Psma4* | Proteasome subunit alpha type-4 | Q9R1P0 | 3.79E−07 | 0.00E+00 | KI only |
| *Oxct1* | Succinyl-CoA:3-ketoacid coenzyme A transferase 1, mitochondrial | Q9D0K2 | 3.72E−07 | 0.00E+00 | KI only |
| *Psmb4* | Proteasome subunit beta type-4 | P99026 | 3.54E−07 | 0.00E+00 | KI only |
| *Tusc5* | Tumour suppressor candidate 5 homolog | Q8C838 | 3.22E−07 | 0.00E+00 | KI only |
| *Map2* | Microtubule-associated protein (Fragment) | G3UZJ2 | 2.41E−07 | 0.00E+00 | KI only |
| *Psmd4* | Isoform Rpn10B of 26S proteasome non-ATPase regulatory subunit 4 | O35226-2 | 2.13E−07 | 0.00E+00 | KI only |
| *Rab1A* | Ras-related protein Rab-1A | P62821 | 1.90E−07 | 0.00E+00 | KI only |
| *Calb2* | Calretinin | Q08331 | 1.83E−07 | 0.00E+00 | KI only |
| *Actr2* | Actin-related protein 2 | P61161 | 1.34E−07 | 0.00E+00 | KI only |
| *Akap12* | Isoform 2 of A-kinase anchor protein 12 | Q9WTQ5-2 | 1.23E−07 | 0.00E+00 | KI only |
| *Gpx4* | Isoform Cytoplasmic of Phospholipid hydroperoxide glutathione oxidase | O70325-2 | 1.15E−07 | 0.00E+00 | KI only |
| *Ahsa1* | Activator of 90-kDa heat-shock protein ATPase homolog 1 | Q8BK64 | 1.12E−07 | 0.00E+00 | KI only |
| *Htatsf1* | HIV Tat-specific factor 1 homolog | Q8BGC0 | 9.15E−08 | 0.00E+00 | KI only |
| *Sart3* | Squamous cell carcinoma antigen recognised by T cells 3 | Q9JLI8 | 8.80E−08 | 0.00E+00 | KI only |
| *Wdr7* | WD repeat-containing protein 7 | Q920I9 | 8.24E−08 | 0.00E+00 | KI only |
| *Rbm10* | Isoform 3 of RNA-binding protein 10 | Q99KG3-3 | 7.90E−08 | 0.00E+00 | KI only |
| *Tkt* | Transketolase | P40142 | 7.83E−08 | 0.00E+00 | KI only |
| *Tmed10* | Transmembrane emp24 domain-containing protein 10 | Q9D1D4 | 7.75E−08 | 0.00E+00 | KI only |
| *Ntm* | Neurotrimin | Q99PJ0 | 7.69E−08 | 0.00E+00 | KI only |
| *Lnp* | Protein lunapark | Q7TQ95 | 7.52E−08 | 0.00E+00 | KI only |

**Table 1** (continued)

| Gene symbol | Protein name | UniProt Acc | Normalised spectral index quantity | | |
|---|---|---|---|---|---|
| | | | KI (Mean) | WT (Mean) | Ratio (KI/WT) |
| *Ddb1* | DNA damage-binding protein 1 | Q3U1J4 | 7.32E−08 | 0.00E+00 | KI only |
| *Mpp2* | Isoform 2 of MAGUK p55 subfamily member 2 | Q9WV34-2 | 6.92E−08 | 0.00E+00 | KI only |
| *Camsap3* | Isoform 2 of calmodulin-regulated spectrin-associated protein 3 | Q80VC9-2 | 6.56E−08 | 0.00E+00 | KI only |
| *Ppp1r9a* | Protein Ppp1r9a | H3BJD0 | 6.25E−08 | 0.00E+00 | KI only |
| *Pccb* | Propionyl-CoA carboxylase beta chain, mitochondrial | Q99MN9 | 6.16E−08 | 0.00E+00 | KI only |
| *Des* | Desmin | P31001 | 5.91E−08 | 0.00E+00 | KI only |
| *Cdh2* | Cadherin-2 | P15116 | 5.88E−08 | 0.00E+00 | KI only |
| *Agpat3* | 1-acyl-sn-glycerol-3-phosphate acyltransferase gamma | Q9D517 | 5.50E−08 | 0.00E+00 | KI only |
| *Ogt* | Isoform 2 of UDP-N-acetylglucosamine | Q8CGY8-2 | 5.42E−08 | 0.00E+00 | KI only |
| *Ahnak* | Protein Ahnak | E9Q616 | 5.34E−08 | 0.00E+00 | KI only |
| *Nfasc* | Neurofascin | Q810U3 | 4.96E−08 | 0.00E+00 | KI only |
| *Mccc2* | Methylcrotonoyl-CoA carboxylase beta chain, mitochondrial | Q3ULD5 | 4.63E−08 | 0.00E+00 | KI only |
| *Mpdz* | Isoform 2 of Multiple PDZ domain protein | Q8VBX6-2 | 3.66E−08 | 0.00E+00 | KI only |
| *Tln1* | Talin-1 | P26039 | 3.32E−08 | 0.00E+00 | KI only |
| *Dsp* | Desmoplakin | E9Q557 | 2.17E−08 | 0.00E+00 | KI only |
| *Slc4a8* | Isoform 2 of Electroneutral sodium bicarbonate exchanger 1 | Q8JZR6-2 | 1.82E−08 | 0.00E+00 | KI only |
| *Slc7a14* | Probable cationic amino acid transporter | Q8BXR1 | 1.07E−08 | 0.00E+00 | KI only |
| *Sgk223* | Tyrosine-protein kinase SgK223 | Q571I4 | 7.46E−09 | 0.00E+00 | KI only |
| Other selected candidates for validation | | | | | |
| *Slc7a5/Lat1* | Large neutral amino acids transporter small subunit 1 | Q9Z127 | 1.66E−07 | 3.11E−08 | 5.34 |
| *Kif5b* | Kinesin-1 heavy chain | Q61768 | 4.04E−07 | 1.00E−07 | 4.04 |
| *Ank3* | Ankyrin-3 (Fragment) | S4R2K9 | 3.06E−07 | 8.73E−08 | 3.51 |
| *Dpysl2/Crmp2* | Dihydropyrimidinase-related protein 2 | O08553 | 7.58E−06 | 2.26E−06 | 3.35 |
| *Gprin1* | G protein-regulated inducer of neurite outgrowth 1 | Q3UNH4 | 3.94E−07 | 2.05E−07 | 1.92 |
| *Syt2* | Synaptotagmin-2 | P46097 | 3.81E−06 | 2.46E−06 | 1.55 |

MGI approved gene symbols and protein names, and UniProt accession numbers are shown. Average of spectral counts from TAP-tagged Na$_V$1.7 knock-in mice (KI) and littermate wild-type control mice (WT) and the ratios from KI group and WT group are displayed.

on Na$_v$1.7 function are predominantly mediated through Crmp2 binding.

It has been shown that the activity of N-type voltage-gated calcium channels (Ca$_V$2.2) in relation to neurotransmitter release from presynaptic terminals in pain pathways is regulated by its protein interactor Crmp2 (Chi *et al*, 2009; Brittain *et al*, 2011). Our study now demonstrates that Crmp2 is also a direct protein interactor of Na$_v$1.7. Therefore, we tested the possibility that Na$_v$1.7 is involved in neurotransmitter release linked to Cav2.2 through Crmp2 using co-IP in an *in vitro* system in HEK cells. However, the results show that Cav2.2 was not immunoprecipitated together with Nav1.7 (Fig 5G), suggesting that Ca$_V$2.2 and Na$_V$1.7 may regulate neurotransmitter release independently.

We further investigated the function of Na$_v$1.7 in relation to presynaptic neurotransmitter release using immunohistochemistry to identify distribution changes in neurotransmitter CGRP and substance P (SP) in the dorsal horn of the spinal cord in Na$_v$1.7

knockout mice (Nassar *et al*, 2004). However, there was no obvious alteration of CGRP or SP levels in lamina I and II in the spinal cord in Na$_V$1.7 knockout mice compared to wild-type littermates (Fig EV2). Thus, levels of neurotransmitters are unaffected by inhibiting release in Na$_V$1.7 null mutants.

### Interactions between Na$_v$1.7 and opioid receptor signalling

Recently a massive potentiation of opioid signalling has been described in sensory neurons of Na$_V$1.7 null mutant mice (Isensee *et al*, 2017). This effect is specific for Na$_V$1.7 as it is not replicated in Na$_V$1.8 null mutant mice. We found that Gprin1, a μ-opioid receptor-binding protein (Ge *et al*, 2009), is also associated with Na$_V$1.7 (Table 1). Furthermore, the protein interaction between Gprin1 and Na$_V$1.7 was confirmed with co-IP *in vitro* (Fig 5D). This suggests close proximity between the sodium channel Na$_V$1.7 and opioid receptors whose efficacy is known to be regulated by sodium.

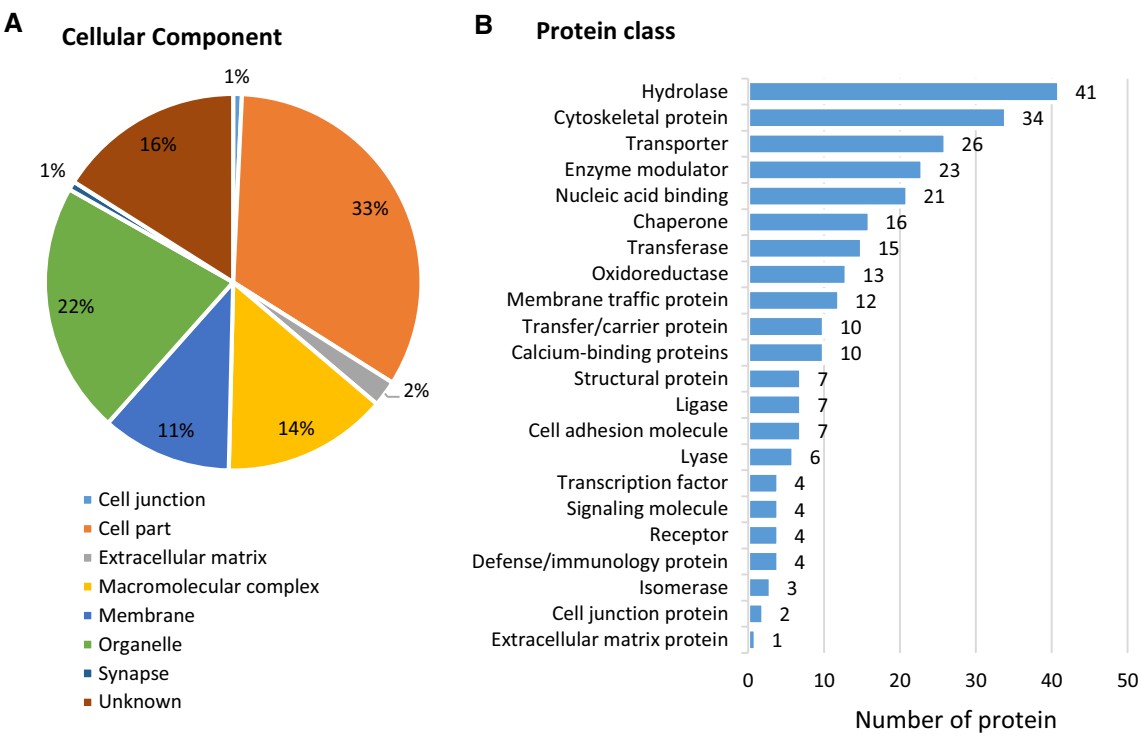

**Figure 6. Analysis of Na$_V$1.7 complex proteins.**

A    Cellular localisation of identified Na$_V$1.7-interacting proteins.

B    Protein class of identified Na$_V$1.7-interacting proteins categorised using PANTHER Classification System.

## Discussion

Experimental evidence has shown that protein–protein interactions play a key role in trafficking, distribution, modulation and stability of ion channels (Shao *et al*, 2009; Catterall, 2010; Leterrier *et al*, 2010; Bao, 2015; Chen-Izu *et al*, 2015; Laedermann *et al*, 2015). Here, we mapped the protein interaction network of Na$_V$1.7 using an AP-MS proteomic approach with an epitope-tagged Na$_V$1.7 knock-in mouse line. This is the first report to define an ion channels' macromolecular complex using an epitope-tagged gene-targeted mouse.

AP-MS requires specific high-affinity antibodies against the target proteins of interest (Wildburger *et al*, 2015). However, binding may compete with normal protein–protein interactions. To overcome these limitations, epitope tags on target proteins were introduced into the AP-MS system. In the last decade, single-step and tandem affinity purification have been widely applied in protein–protein interaction studies (Fernandez *et al*, 2009; Wildburger *et al*, 2015). In contrast to ss-AP, TAP produces lower background and less contamination. However, due to its longer experimental washing procedures and two-step purification, TAP coupled with MS analysis may not be sufficiently sensitive to detect transient and dynamic protein–protein interactions. In recent years, along with newly developed highly sensitive mass spectrometer techniques and powerful quantitative proteomics analysis methods, ss-AP was employed to identify both transient and stable protein–protein interactors (Oeffinger, 2012; Keilhauer *et al*, 2015). For example, using a single-step FLAG approach, Chen and colleagues defined specific novel interactors for the catalytic subunit of PP4, which they had not previously observed with TAP-MS (Chen & Gingras, 2007). Thus, ss-AP followed by sensitive LC-MS/MS analysis was applied in this study. In fact, many dynamic modulator proteins were identified to interact with Na$_V$1.7 in this study, such as calmodulin (Calm1; Table EV1), which was found to bind to the C-terminus of other VGSCs Na$_V$1.4 and Na$_V$1.6, thereby regulating channel function (Herzog *et al*, 2003). Apart from the sodium channel β3 subunit and the known Na$_V$1.7 protein interactor Crmp2, a broad range of important novel interactors that belong to different protein classes, such as cytoskeletal/structural/cell-adhesion proteins and vesicular/trafficking/transport proteins (Fig 6B), have been identified in this study. Four proteins, Crmp2, Nedd4-2, FGF13 and Pdzd2, have previously been reported as Na$_V$1.7 protein interactors (Sheets *et al*, 2006; Shao *et al*, 2009; Ho *et al*, 2012; Yang *et al*, 2017). Laedermann *et al* (2013) showed that the E3 ubiquitin ligase Nedd4-2 regulates Na$_V$1.7 by ubiquitinylation in the pathogenesis of neuropathic pain. Shao *et al* (2009) demonstrated that Pdzd2 binds to the second intracellular loop of Na$_V$1.7 by a GST pull-down assay in an *in vitro* system. Recently, Zhang's group revealed that FGF13 regulates heat nociception by interacting with Na$_V$1.7 (Yang *et al*, 2017). We did not find the previously reported Na$_V$1.7 interactors Nedd4-2, FGF13 or Pdzd2. This may be because Nedd4-2 and FGF13 only bind to Na$_V$1.7 in neuropathic pain conditions and in heat nociception, respectively, and Pdzd2 shows strong binding *in vitro* but not *in vivo*.

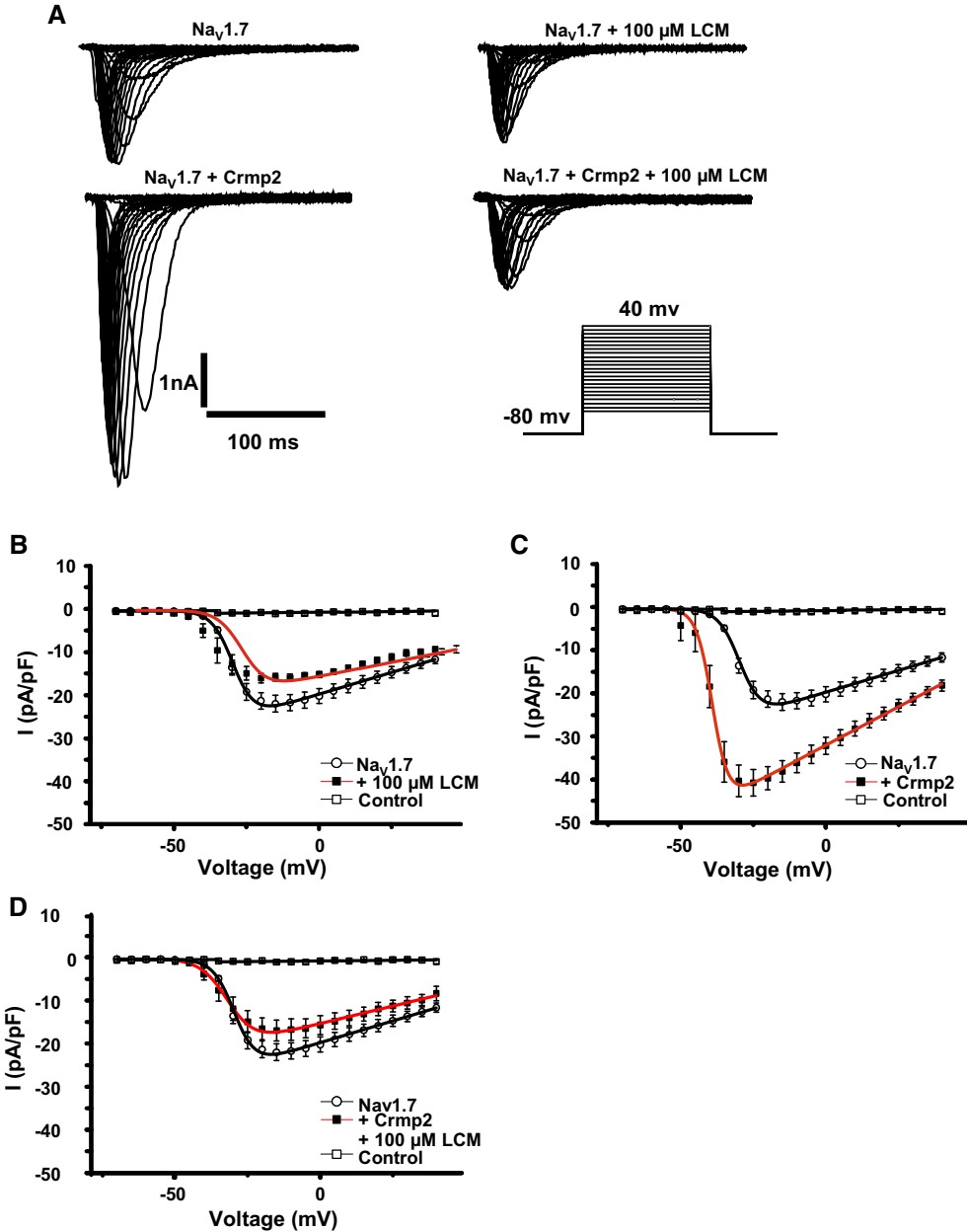

**Figure 7. Electrophysiological characterisation of Na$_V$1.7 in HEK293 cells following transfection with Crmp2 and incubation with lacosamide (LCM).**

A   Representative raw current traces of Na$_V$1.7 stably expressed in HEK293 cells in response to the activation pulse protocol shown. Each trace shows a different condition: + Crmp2 transfection and + incubation with 100 μM LCM.

B   IV plot of Na$_V$1.7 current density in the absence and presence of 100 μM LCM. Compared with Na$_V$1.7 basal currents ($n$ = 10), LCM incubation had no significant effect on sodium channel density ($n$ = 12, $P$ = 0.402).

C   IV plot of Na$_V$1.7 in HEK293 cells in the presence and absence of Crmp2 transfection. Compared to Na$_V$1.7 basal currents, Crmp2 transfection caused a significant increase in Na$_V$1.7 current density ($n$ = 16, $P$ = 0.0041).

D   IV plot showing current density of Na$_V$1.7 following transfection of Crmp2 and incubation with 100 μM LCM. Incubation with LCM reversed the Crmp2-mediated current increase ($n$ = 10, $P$ = 0.0442).

Data information: Data were analysed using one-way ANOVA with Tukey's *post hoc* test. All data are mean ± SEM.

Scn3b and Crmp2 have previously been proposed to associate with Na$_V$1.7 (Ho *et al*, 2012; Dustrude *et al*, 2013). We demonstrated the physical interaction between these proteins and Na$_V$1.7 by co-IP. Furthermore, we demonstrated that transient overexpression of Crmp2 can upregulate Na$_V$1.7 current density in stably expressing Na$_V$1.7 HEK293 cells and that this upregulation can be reversed by applying lacosamide. Previous studies reported that no change in Na$_V$1.7 currents with Crmp2 overexpression in CAD cells (Wang *et al*, 2010; Dustrude *et al*, 2013). This could be attributed perhaps to the different species' Crmp2 or the different transfection

conditions. Taken together, FLAG ss-AP coupled with quantitative MS seems to be a powerful and reliable tool for investigating protein interactions of membrane ion channels.

Using co-IP and a co-expression in an *in vitro* system, we also confirmed the direct physical interaction between Na$_V$1.7 and novel protein interactors of interest including Syt2, Lat1, Tmed10 and Gprin1. Synaptotagmin is a synaptic vesicle membrane protein that functions as a calcium sensor for neurotransmission (Chapman, 2002). Sampo *et al* (2000) showed a direct physical binding of synaptotagmin-1 with Na$_V$1.2 at a site, which is highly conserved across all voltage-gated sodium channels suggesting that the synaptotagmin family can associate with other VGSCs. Na$_V$1.7 found at presynaptic terminals appears to be involved in neurotransmitter release (Weiss *et al*, 2011; Black *et al*, 2012; Minett *et al*, 2012). Our results revealed the physical interaction between Syt2 and Na$_V$1.7. Interestingly, other synaptic proteins also coprecipitated with Na$_v$1.7 including SNARE complex protein syntaxin-12 (Table EV1). These data suggest that Na$_V$1.7 may regulate neurotransmitter release through Syt2 and syntaxin-12 in the peripheral and central terminals of the spinal cord.

Gabapentin was developed to treat epilepsy, but it is now used to treat various forms of chronic pain. However, the analgesic mechanisms of gabapentin are not entirely clear, but involve the inhibition of the voltage-gated calcium channel Ca$_V$2.2 by blocking the trafficking of α2δ-1- and α2δ-1-mediated trafficking of the Ca$_V$2.2 channel complex to reduce channel activity in DRG neurons (Hendrich *et al*, 2008; Cassidy *et al*, 2014). As the transporter of gabapentin, Lat1 has been identified as a Na$_V$1.7 protein interactor, and it is conceivable that Lat1 may also be involved in the regulation of Na$_V$1.7 channel function. However, future experiments are required to determine the functional significance of the Na$_v$1.7-Lat1 interaction in this regard. As a member of the p24 family, Tmed10 was selected due to its well-characterised properties as a protein trafficking regulator. Previous studies have highlighted that Tmed10 plays an important role as a cargo receptor in the trafficking of various membrane proteins. For example, Tmed10 has been observed to modulate the transport and trafficking of amyloid-β precursor protein (Vetrivel *et al*, 2007), endogenous glycosylphosphatidylinositol (GPI)-anchored proteins CD59 and folate receptor alpha (Bonnon *et al*, 2010), and several G protein-coupled receptors (GPCRs) (Luo *et al*, 2011). Tmed10 was confirmed as a Na$_V$1.7 protein interactor in this study, suggesting Tmed10 may regulate Na$_V$1.7 trafficking. The underlying mechanisms need to be further investigated.

VGSCs are known to exist in macromolecular complexes (Meadows & Isom, 2005). The β subunits are members of the immunoglobulin (Ig) domain family of cell-adhesion molecules (CAM). As well as sodium channel α subunits, the β subunits also bind to a variety of cell-adhesion molecules such as neurofascin, contactin, tenascins and NrCAMs (Srinivasan *et al*, 1998; Ratcliffe *et al*, 2001; McEwen & Isom, 2004; Cusdin *et al*, 2008; Namadurai *et al*, 2015). In our data set (Table EV1), the sodium channel β3 subunit and some CAMs, such as Ncam1 and neurofascin, have been found to associate with the Na$_V$1.7 α subunit. The crystal structure of the human β3 subunit has been solved recently. The β3 subunit Ig-domain assembles as a trimer in the crystal asymmetric unit (Namadurai *et al*, 2014). This raises the possibility that trimeric β3 subunits binding to Na$_V$1.7 α subunit(s) form a large complex together with other sodium channels, as well as with CAMs and cytoskeletal proteins in the plasma membrane.

Na$_V$1.7 has also been linked to opioid peptide expression, and enhanced activity of opioid receptors is found in the Na$_V$1.7 null mutant mouse (Minett *et al*, 2015). Interestingly, Gprin1, which is known to interact with opioid receptors as well as other GPCRs, was found to co-immunoprecipitate with Na$_V$1.7. This suggests that GPCR sodium channel interactions could add another level of regulatory activity to the expression of Na$_V$1.7. Intriguingly, opioid receptors are well known to be regulated by sodium (Ott *et al*, 1988; Fenalti *et al*, 2014). Deleting Na$_V$1.8 had no effect on μ-opioid receptor efficacy measured with fentanyl, whilst Na$_V$1.7 deletion potentiated opioid action substantially (preprint: Kanellopoulos *et al*, 2017). Thus, the proximity of Na$_V$1.7 and μ-opioid receptors mediated by Gprin1 may contribute to this regulation.

More recently, Branco and colleagues reported that Na$_V$1.7 in hypothalamic neurons plays an important role in body weight control (Branco *et al*, 2016). We found that Na$_V$1.7 was not only present in the arcuate nucleus but also in other regions of the brain such as the medial amygdala, medial habenula, anterodorsal thalamic nucleus, laterodorsal thalamic nucleus, and in the subfornical organ, substantia nigra reticular part and the red nucleus magnocellular part of the midbrain, and in neurons of the pontine nuclei located in the hindbrain. Na$_V$1.7 thus has other functions in the CNS that remain to be elucidated.

Overall, the present findings provide new insights into the interactome of Na$_V$1.7 for advancing our understanding of Na$_V$1.7 function. Our data also show that the ss-AP-coupled LC-MS/MS is a sensitive, reliable and high-throughput approach to identify protein–protein interactors for membrane ion channels, using epitope-tagged gene-targeted mice.

# Materials and Methods

### Generation of a TAP-tagged Na$_V$1.7-expressing stable HEK293 cell line

A HEK293 cell line stably expressing TAP-tagged Na$_V$1.7 was established as previously described (Koenig *et al*, 2015). Briefly, a sequence encoding a TAP-tag (peptide: SRK DHL IHN VHK EEH AHA HNK IEN LYF QGE LPT AAD YKD HDG DYK DHD IDY KDD DDK) was inserted immediately prior to the stop codon of Na$_V$1.7 in the SCN9A mammalian expression construct FLB (Cox *et al*, 2006). The TAP-tag at the extreme C-terminus of Na$_V$1.7 comprises a HAT domain and 3 FLAG-tags, enabling immunodetection with either anti-HAT or anti-FLAG antibodies. The function and expression of TAP-tagged Na$_V$1.7 in this HEK293 cell line were characterised with both immunocytochemistry and electrophysiological patch clamp analysis.

### Generation of Na$_V$1.7$^{TAP}$ knock-in mice

The gene targeting vector was generated using a BAC homologous recombineering-based method (Liu *et al*, 2003). Four steps were involved in this procedure (Fig EV1). Step 1, two short homology arms (HA) HA3 and HA4 corresponding to 509-bp and 589-bp sequences within intron 26 and after exon 27 of Na$_V$1.7, respectively, were amplified by PCR using a BAC bMQ277g11 (Source Bioscience, Cambridge, UK) DNA as a template, and then inserted into a retrieval vector pTargeter (Fernandez *et al*, 2009; gift

from Dr. Seth GN Grant) by subcloning. Step 2, a 9.1-kb genomic DNA fragment (3.4 kb plus 5.8 kb) was retrieved through homologous recombineering by transforming the *Kpn*I-linearised pTargeter-HA3-HA4 vector into EL250 *E. coli* cells containing BAC bMQ277g11. Step 3, short homology arms HA1 and HA2 corresponding to 550 bp (before stop codon of Na$_V$1.7) and 509 bp (starting from stop codon of Na$_V$1.7), respectively, were amplified by PCR, and then cloned into pneoflox vector (Fernandez *et al*, 2009; gift from Dr. Seth GN Grant) containing the TAP-tag, leaving inbetween the TAP-tag sequence, 2 LoxP sites, PGK and EM7 promoters, the G418$^r$ gene and a SV40 polyadenylation site. Step 4, the cassette flanked by two homology arms (HA1 and HA2) was excised by *Xho*I and *Bgl*II digestion and transformed into recombination-competent EL250 cells containing the pTargeter-HA3-HA4 plasmid. Then, the TAP tag cassette was inserted into the pTargeter-HA3-HA4 vector by recombination in EL250 *E. coli* cells. The correct recombination and insertion of the targeting cassette were confirmed by restriction mapping and DNA sequencing. The complete gene targeting vector containing the 5′-end homologous Na$_V$1.7 sequence of 3.4 kb and a 3′-end homology arm of 5.8 kb was linearised with *Pme*I digestion for ES cell electroporation. All the homology arms HA1, HA2, HA3 and HA4 were amplified with NEB Phusion PCR Kit using bMQ277g111 BAC clone DNA as a template. Primers used to create the recombination arms included:

HA1XhoIF (HA1, forward)— acactcgagAGCCAAACAAAGTCCAGCT
HA1XbaIR (HA1, reverse)—tgttctagaTTTCCTGCTTTCGTCTTTCTC
HA2Acc65IF (HA2, forward)—tgaggtacctagAGCTTCGGTTTTGATACACT
HA2BglIIR (HA2, reverse)—gatagatctTTGATTTTGATGCAATGTAGGA
HA3SpeIF (HA3, forward)—ctcactagtCTCTTCATACCCAACATGCCTA
HA3KpnIR (HA3, reverse)—aatggtaccGGATGGTCTGGGACTCCATA
HA4KpnIF (HA4, forward)—gaaggtaccGCTAAGGGGTCCCAAATTGT
HA4PmeIR (HA4, reverse)—tcagtttaaacGGGATGGGAGATTACGAGGT

To generate TAP-tagged Na$_V$1.7 mice, the linearised targeting vector was transfected into 129/Sv ES cells. Cells resistant to G418 were selected by culturing for 9 days. Recombined ES cell clones were identified using Southern blot-based screening. Three clones that were confirmed to be correct using Southern blot were injected into C57BL/6 blastocysts at the Transgenic Mouse Core facility of the Institute of Child Heath (ICH). The chimeric animals were crossed to C57BL/6, and the germline transmission was confirmed by Southern blot. The neomycin cassette was removed by crossing with global Cre mice. The correct removal of the neomycin cassette and TAP-tag insertion was confirmed by Southern blot, genotyping (PCR) and RT–PCR.

The genomic DNA was extracted from ear punches and genotyping analysis was achieved using a standard PCR. Primers used for PCR included:

5aF1 (forward)—ACAGCCTCTACCATCTCTCCACC
3aR4 (reverse)—AACACGAGTGAGTCACCTTCGC

The wild-type Na$_V$1.7 allele and TAP-tagged Na$_V$1.7 allele gave a 170-bp band and a 411-bp band, respectively. The TAP-tagged Na$_V$1.7 mRNA was confirmed by RT–PCR. Briefly, total RNA was extracted from dorsal root ganglia with Qiagen RNEasy kit (Qiagen) and 1.0 μg was used to synthesise cDNA using Bio-Rad iScript cDNA synthesis kit with oligo-dT primers. The following primers were used to detect Na$_V$1.7 mRNA:

E25-26-F (forward)—CCGAGGCCAGGGAACAAATTCC
3′UTR-R (reverse)—GCCTGCGAAGGTGACTCACTCGTG

The wild-type and TAP-tagged Na$_V$1.7 alleles gave a 1,521-bp band and a 1,723-bp band, respectively.

## Immunocytochemistry

TAP-tagged Na$_V$1.7-HEK293 cells and their parental HEK293 cells were plated on poly-D-lysine coated coverslips in 24-well plates and cultured at 37°C/5% CO$_2$ in DMEM supplemented with 10% foetal bovine serum (Life Technologies), 50 U/ml penicillin, 50 μg/ml streptomycin and 0.2 mg/ml G418 (only for TAP-tagged Na$_V$1.7-HEK293 cells); 24 h later, cells were fixed in cooled methanol at −20°C for 10 min and then permeabilised with cooled acetone at −20°C for 1 min. After three washes with 1× PBS, the cells were incubated with blocking buffer containing 1× PBS, 0.3% Triton X-100 and 10% goat serum at room temperature for 30 min. Then, the fixed cells were incubated with anti-FLAG antibody (1:500 in blocking buffer, F1804, Sigma) at 4°C overnight. After 3 washes with 1× PBS, cells were incubated with secondary antibody goat anti-mouse IgG conjugated with Alexa Fluor 488 (A11017, Invitrogen) at room temperature for 2 h. Then, coverslips were washed 3 times in 1× PBS and cells were mounted with VECTASHIELD HardSet Antifade Mounting Medium containing DAPI (H-1400, Vectorlabs) and visualised using a fluorescence microscope (Leica).

## Immunohistochemistry

Following anaesthesia, mice were transcardially perfused with PBS, followed by 1% (for brain) or 4% (for spinal cord) paraformaldehyde in PBS. Brains and spinal cord were dissected and incubated in fixative for 4 h at 4°C, followed by 30% sucrose in PBS for 2 days at 4°C. Tissue was embedded in O.C.T. (Tissue-Tek) and snap-frozen in a dry ice/2-methylbutane bath. Brain coronal and spinal cord cross cryosections (20 μm) were collected on glass slides (Superfrost Plus, Polyscience) and stored at −80°C until further processing. For FLAG-tag immunohistochemistry in the brain, sections were incubated in blocking solution (4% horse serum, 0.3% Triton X-100 in PBS) for 1 h at room temperature, followed by incubation in mouse anti-FLAG antibody (F-1804, Sigma) diluted 1:200 in blocking solution for 2 days at 4°C. Following three washes in PBS, bound antibody was visualised using either an Alexa 488- or 594-conjugated goat anti-mouse secondary antibody (1:800, Invitrogen). To facilitate the identification of brain regions and the corresponding bregma levels, nuclei were counterstained with Hoechst 33342 (1:10,000, Invitrogen). Sections were mounted with DAKO fluorescence mounting medium. Fluorescence images were acquired on an epifluorescence microscope (BX61 attached to a DP71 camera, Olympus) or a confocal laser scanning microscope (LSM 780, Zeiss). Images were assembled and minimally adjusted in brightness and contrast using Adobe Photoshop Elements 10. Bregma levels and brain regions were identified according to the stereotaxic coordinates in "The Mouse Brain" atlas by Paxinos and Franklin (2001). The immunohistochemistry experiments in spinal cord were performed as described previously (Zhao *et al*, 2010) using the following primary antibodies: anti-FLAG antibody (1:400; F1804, Sigma), anti-CGRP antibody (1:100; #24112, ImmunoStar), anti-substance P

(1:100; #20064, ImmunoStar) and biotin-conjugated isolectin B4 (1:200; L2140, Sigma).

## Behavioural analysis

All behavioural tests were approved by the United Kingdom Home Office Animals (Scientific Procedures) Act 1986. Age (6–12 weeks)-matched KI mice (four males and three females) and littermate wild-type (WT) controls (three males and four females) were used for acute pain behaviour studies. The experimenters were blind to the genetic status of test animals. The Rotarod, Hargreaves', von Frey and Randall–Selitto tests were performed as described (Zhao *et al*, 2006). We used the same set of mice to perform acute behavioural tests. The order was Rotarod, von Frey, Hargreaves and Randall–Selitto tests. We left one-day gap between Rotarod, von Frey and Hargreaves test, and 3-day gap between Hargreaves test and Randall–Selitto test. The formalin test was carried by intraplantar injections of 20 μl of 5% formalin. Age (8–12 weeks)-matched seven male KI mice and 10 male littermate wild-type (WT) controls were used. The mice were observed for 45 min and the time spent biting and licking the injected paw was monitored and counted. Two phases were categorised, the first phase lasting 0–10 min and the second phase 10–45 min. All data are presented as mean ± SEM.

## Single-step and tandem affinity purification

In each round of sample preparation for single-step and tandem affinity purification, DRG, olfactory bulbs, spinal cord and hypothalamus samples were homogenised using Precellys Tissue Homogenizer and Precellys lysis kit (Precellys ceramic kit 1.4 mm, Order no. 91-PCS-CK14, Peqlab) in 1% CHAPS lysis buffer (30 mM Tris–HCl pH 7.5, 150 mM NaCl, 1% CHAPS, 1 complete EDTA-free protease inhibitor cocktail (Roche) in 10 ml of CHAPS lysis buffer) and further homogenised using an insulin syringe. The lysates were incubated shaking horizontally on ice and clarified by centrifugation at 14,000 *g* for 8 min at 4°C. Protein concentrations were measured using the Pierce BCA Protein Assay Kit (Product no. 23225, Thermofisher), and a total starting amount of 10 mg of protein containing supernatant was incubated with anti-FLAG M2 Magnetic beads (M8823, Sigma). The coupling was carried out for 2 h at 4°C using an end-over-end shaker. Magnetic beads were collected on a DynaMAG rack (Invitrogen) and washed three times in 1% CHAPS Buffer and 1× AcTEV protease cleavage buffer (50 mM Tris–HCl pH 8.0, 0.5 mM EDTA, 1 mM DTT). Bead-captured Na$_V$1.7 TAP-tag complex was released from the beads by incubation with AcTEV protease (#12575015, Invitrogen) at 30°C for 3 h, finalising the single-step purification. For the tandem affinity purification, protein eluates were collected after AcTEV cleavage and 15× diluted in protein binding buffer (50 mM sodium phosphate, 300 mM NaCl, 10 mM imidazole, 0.01% Tween 20; pH 8.0). AcTEV-cleaved Na$_V$1.7 and its complex were then captured using Ni-NTA beads (#36111, Qiagen). Ni-NTA beads were washed three times in protein binding buffer and incubated on an end-over-end shaker overnight at 4°C. TAP-tagged Na$_V$1.7 protein complexes were released from the Ni-NTA beads by boiling in 1× SDS protein sample buffer.

## Western blot

Proteins for Western blot were isolated from freshly excised different tissues, such as DRG, spinal cord, sciatic nerve, olfactory bulb, cortex, hypothalamus, cerebellum, skin, lung, heart and pancreas taken from TAP-tagged Na$_V$1.7 mice and littermate control mice. The protein samples were prepared the same as described in the section of single-step and tandem affinity purification in Materials and Methods. Briefly, proteins extracted from different tissues were homogenised in 1% CHAPS lysis buffer.

The nuclear fraction and cell debris were removed by centrifugation at ∼20,000 *g* for 15 min at 4°C. Protein concentrations were determined with Pierce BCA protein assay kit, and then samples of 40 μg were separated on SDS–PAGE gel in Bio-Rad Mini-PROTEAN Vertical Electrophoresis Cell System and blotted to the Immobilin-P membrane (IPVH00010, Millipore) in transfer buffer (25 mM Tris–HCl, pH 8.3, 192 mM glycine, 0.1% SDS and 20% methanol) for 1 h at 100 V with a Bio-Rad transfer cell system. The membrane was blocked in blocking buffer [5% nonfat milk in PBS–Tween buffer (0.1% Tween 20 in 1× PBS)] for 1 h at room temperature and then incubated with primary antibody anti-FLAG (1:1,000; Sigma, catalog #F1804) and anti-HAT (1:400; LSBio, #LS-C51508) in blocking buffer overnight at 4°C. The membrane was washed three times with TBS–Tween (20 mM Tris, 150 mM NaCl, 0.1% Tween 20, pH 7.5) and then incubated with secondary antibody goat anti-mouse or goat anti-rabbit IgG-HRP (1:4,000; Jackson Immuno-Research Laboratories) in TBS–Tween at room temperature for 2 h. Detection was performed using a Western Lightning Chemiluminescence Reagent (Super Signal Western Dura, Thermo Scientific, #34075) and exposed to BioMax film (Kodak). Other primary antibodies used for Western blotting were ankyrin-3 (Santa Cruz, SC-12719), neurotrimin (Santa Cruz, SC-390941), Kif5b (Santa Cruz, SC-28538), PEBP1 (Thermo fisher, 36-0700), neurofascin (Abcam, ab31457), neuroligin (Abcam, ab177107) and AKAP12 (Abcam, ab49849). Concentrations were applied as suggested by the manufacturers.

## Plasmids

The following plasmids were obtained from Addgene (Cambridge, MA): Tmed10 (TMED-BIO-HIS, #51852), Lat1 (pEMS1229, #29115). Gprin1 plasmid was obtained from OriGene (#RC207340). In order to perform *in vitro* validation of Na$_V$1.7 interaction with synaptotagmin-2, we first cloned the human gene for insertion into a mammalian expression plasmid. Synaptotagmin-2 was cloned from human dorsal root ganglion neuronal tissue. Whole mRNA was first reverse-transcribed into a cDNA library for PCR. Following insertion into a TOPO vector, this was then used as the template to create the synaptotagmin-2 insert, which was then successfully cloned into a pcDNA3.1 IRES-AcGFP plasmid using the Gibson assembly method. The collapsin Response Mediator Protein 2 (Crmp2) gene was cloned from human dorsal root ganglia neuron mRNA and cloned into a pcDNA3.1 plasmid using Gibson assembly. The Scn3b mammalian-expressing vector was used to investigate loss of function of Na$_V$1.7 (Cox *et al*, 2006). The HA-tagged Ca$_V$2.2 was a gift from Prof. Annette Dolphin, and was co-expressed with the auxiliary subunits, alpha2delta-1 and beta1b.

## Co-immunoprecipitation

Mouse tissues used in Co-IP experiments were chosen on the basis of its known Na$_V$1.7 expression in olfactory bulb, hypothalamus, spinal cord (lumbar enlargement), dorsal root ganglia (all cervical, thoracic, lumbar and sacral DRGs) and sciatic nerve (both sides). All tissues from individual mice were either pooled or treated as individual tissue samples depending on experimental requirements. Tissues were flash-frozen in dry ice immediately following dissection and stored at −80°C to avoid protein degradation. HEK293 cells stably expressing TAP-tagged Na$_V$1.7 were harvested by trypsinisation followed by centrifugation at 800 rpm for 5 min. Cell pellets were stored at −80°C to avoid protein degradation. Samples were lysed in a 1% CHAPS lysis buffer containing a protease inhibitor cocktail and homogenised using ceramic zirconium oxide mix beads of 1.4 and 2.8 mm lysing kit and homogeniser (Precellys). A total of 10–25 mg of protein was incubated with anti-FLAG M2 magnetic beads for 2 h at 4°C. The bead–protein complex was then washed with three cycles of five resin volumes of 1% CHAPS buffer and once with TEV–protease buffer (Invitrogen). The tagged protein was cleaved from the beads by the addition of TEV protease enzyme (Invitrogen) and incubated for 3 h at 37°C to elute the protein complex. Sample eluate was then separated from the beads and stored at −80°C until used for either Western blotting or mass spectrometry. Co-immunoprecipitation in HEK293 cells was performed after transfection of the construct with the gene of interest into the TAP-tagged Na$_V$1.7 stable cell line using a standard lipofectamine protocol (Lipofectamine 200, Invitrogen, #52887). All cells were left 48 h after transfection before use in experiments.

## Liquid chromatography–tandem mass spectrometry analysis

Proteins cleaved from anti-FLAG M2 magnetic beads after affinity purification were tryptic-digested following a FASP protocol (Wisniewski *et al*, 2009). In brief, proteins were loaded to 30-kDa filters (Millipore) and then filter units were centrifuged at 14,000 *g* for 15 min to remove other detergents. Two hundred μl of urea buffer (10 mM dithiothreitol 8 M urea (Sigma) in 0.1 M Tris–HCl, pH 8.5) was added to the filters and left at room temperature for 1 h to reduce proteins. Then, filters were centrifuged to remove dithiothreitol. Two hundred μl of 50 mM iodoacetamide (IAA) in urea buffer was added to filters and left for 30 min in the dark. Filters were centrifuged as before to remove IAA. Then, the samples were buffer exchanged twice using 200 μl of urea buffer, and one more time using 200 μl of 50 mM NH$_4$HCO$_3$ in water. Forty μl of 50 ng/μl trypsin in 50 mM NH$_4$HCO$_3$ was added to filter, filters were vortexed briefly, and proteins were digested at 37°C for overnight. After tryptic digestion, the filters were transferred to new collection tubes and the peptides collected by placing the filter upside down and spinning. The samples were acidified with CF$_3$COOH and desalted with C18 cartridge (Waters). The pure peptides were dried by Speedvac (Millipore) and resuspended with 20 μl of 2% ACN, 0.1% FA. Five μl of samples was injected into Orbitrap Velos mass spectrometry (Thermo) coupled to a UPLC (Waters; Thézénas *et al*, 2013).

Analysis was carried out by nano-ultra-performance liquid chromatography–tandem MS (nano-UPLC-MS/MS) using a 75 μm inner diameter × 25 cm C18 nanoAcquity UPLC™ column (1.7 μm particle size, Waters) with a 180-min gradient of 3–40% solvent B (solvent A: 99.9% H$_2$O, 0.1% formic acid; solvent B: 99.9% ACN, 0.1% formic acid). The Waters nanoAcquity UPLC system (final flow rate, 250 nl/min) was coupled to a LTQ Orbitrap Velos (Thermo Scientific, USA) run in positive ion mode. The MS survey scan was performed in the FT cell recoding a window between 300 and 2,000 m/z. The resolution was set to 30,000. Maximum of 20 MS/MS scans were triggered per MS scan. The lock mass option was enabled, and polysiloxane (m/z 371.10124) was used for internal recalibration of the mass spectra. CID was done with a target value of 30,000 in the linear ion trap. The samples were measured with the MS setting charge state rejection enabled, and only more than 1 charges procures ions selected for fragmentation. All raw MS data were processed to generate MGF files (200 most intense peaks) using the Proteowizard v.2.1.2476 software. The identification of proteins was performed using MGF files with the central proteomics facilities pipeline. *Mus musculus* (Mouse) database containing entries from UniProtKB was used in CPF Proteomics pipeline for data analysis. This pipeline combines database search results from three search engines (Mascot, OMSSA and X!tandem k-score). The search was carried out using the following parameters. Trypsin was the enzyme used for the digestion of the proteins, and only one missed cleavage was allowed. The accepted tolerance for the precursor was 20 ppm and 0.5 Da for the fragment. The search encompassed 1+, 2+ and 3+ charge state, fixed modification for cysteine carbamidomethyl and variable modification for asparagine and glutamine deamidation, and methionine oxidation. All trypsin fragments were added to an exclusion list. False discovery rate was calculated by peptide/proteinprophet or estimated empirically from decoy hits a 1% FDR cut-off was used to filter identified proteins. The label-free analysis was carried out using the normalised spectral index (SINQ; Trudgian *et al*, 2011). The mass spectrometry proteomics data have been deposited to the ProteomeXchange Consortium (http://www.proteomexchange.org/) via the PRIDE (Vizcaíno *et al*, 2016) partner repository with the dataset identifier PXD004926.

## Electrophysiology and patch clamp recordings

Whole-cell patch clamp recordings were conducted at room temperature (21°C) using an AxoPatch 200B amplifier and a Digidata 1322A digitiser (Axon Instruments), controlled by Clampex software (version 10, Molecular Devices). Filamented borosilicate microelectrodes (GC150TF-10, Harvard Apparatus) were pulled on a Model P-97 Flaming/Brown micropipette puller (Sutter Instruments) and fire-polished to a resistance of between 2.5 and 4 MOhm. Standard pipette intracellular solution contained: 10 mM NaCl, 140 mM CsF, 1.1 mM EGTA, 1 mM MgCl$_2$ and 10 mM HEPES. The standard bathing extracellular solution contained: 140 mM NaCl, 1 mM MgCl$_2$, 3 mM KCl, 1 mM CaCl$_2$ and 10 mM HEPES. Both intracellular and extracellular solutions were adjusted to a physiological pH of 7.3. The amplifier's offset potential was zeroed when the electrode was placed in the solution. After a giga-seal was obtained, short suction was used to establish whole-cell recording configuration. Errors in series resistance were compensated by 70–75%. Cells were briefly washed with extracellular solution before a final 2 ml of solution was transferred to the dish. Cells were held at −100 mV for 2 min before experimental protocols were initiated. Currents were elicited by 50-ms depolarisation

steps from −80 mV to +80 mV in 5 mV increments. Compounds were added and mixed at the desired concentrations in extracellular solution before being added to the bath. Following addition of the compound, protocols were repeated on previously unrecorded cells. All currents were leak-subtracted using a p/4 protocol. The following compounds were used in electrophysiology experiments: lacosamide ((R)-2-acetamido-N-benzyl-3-methoxypropionamide) was obtained from Toronto Research Chemicals Inc (L098500) and tetrodotoxin was obtained from Sigma-Aldrich (T8024). Incubation with lacosamide was done for 5 h prior to recording.

Voltage-clamp experiments were analysed using pCLAMP software and Origin (OriginLab Corp., Northampton, MA) software programs. Current density–voltage analysis was carried out by measuring peak currents at different applied voltage steps and normalised to cell capacitance (pA/pF). Voltage-dependent activation data were fitted to a Boltzmann equation y = (A2 + (A1 − A2)/ (1 + exp((Vh − x)/k)))*(x − Vrev), where A1 is the maximal amplitude, Vh is the potential of half-maximal activation, x is the clamped membrane potential, Vrev is the reversal potential and k is a constant. All Boltzmann equations were fitted using ORIGIN software.

### Na$_V$1.7 interaction protein selection and function analysis

Candidate proteins that may interact with Nav1.7 were selected by two criteria: (i) present in at least two knock-in biological experiments but absent from wild-type experiments; (ii) present in more than three knock-in and more than one wild-type experiments, the ratio of average abundance is more than 1.5-fold increased in knock-in experiments as compared with wild-type experiments. Further cellular component and function classification were performed on PANTHER Classification System (11.0). Ingenuity pathway analysis (IPA) (QIAGEN) was used to elucidate pathways and protein interaction networks using candidate proteins.

### Southern blot analysis

The genomic DNA was extracted from either ES cells or tails of mice following the procedures as described (Sambrook & Russell, 2001). The probes for Southern blot were amplified by PCR using mouse genomic DNA isolated from C57BL/6 as a template and purified with a Qiagen Gel Purification Kit. The restriction enzymes *Stu*I, *Bsp*HI and *Psi*I were used to digest genomic DNA for either wild-type and knock-in bands. The sizes of wild-type and knock-in bands are shown in Fig 2B. The primers used to create probes (5′ external probe: 768 bp; 3′ external probe: 629 bp) included:
5′PF (5′ probe, forward)—ACCAAGCTTTTGATATCACCATCAT
5′PR (5′ probe, reverse)—CAACTCGAGAACAGGTAAGACATGA CAGTG
3′PF (3′ probe, forward)—TTTAAGCTTCCTGCCCCTATTCCTGCT
3′PR (5′ probe, reverse)—TTAGGATCCATGCACTACTGACTTGCT TATAGGT

### Statistical analysis

Statistical analysis was performed using either repeated-measures ANOVA with Bonferroni *post hoc* testing or unpaired Student's *t*-test

as described in the results or figure legends. The GraphPad Prism 6.0 was used to perform the statistical analysis. All data are presented as mean ± SEM, and significance was determined at *P* < 0.05.

**Expanded View** for this article is available online.

### Acknowledgments

We are grateful to Professor Seth GN Grant and Dr. Esperanza Fernandez for the TAP-tag plasmids. We thank the Mass Spectrometry Laboratory of Target Discovery Institute at University of Oxford for performing the mass spectrometry analysis. We thank Dr. Massimo Signore who performed electroporation and blastocysts microinjection of ES cells in JP Martinez-Barbera's research group, ICH, UCL. This study was supported by the Wellcome Collaborative Award 200183/Z/15/Z (to J.W., J.Z., J.C. and S.G.) and Investigator Award 101054/Z/13/Z (to J.W.), the Medical Research Council (MRC) Grant G091905 (to J.W.) and MRC CDA G1100340 (to J.C.), the Deutsche Forschungsgemeinschaft (DFG) Grants SFB 894/A17 (to F.Z.) and PY90/1-1 (to M.P.).

## Author contributions

JZ generated the TAP-tagged Na$_V$1.7 knock-in mouse. QM, SJG and JZ maintained the mouse line. JZ and QM contributed to pain behavioural study and analysed the data. JZ, MJ and JK generated the TAP-tagged Na$_V$1.7 HEK293 cell line. AHK and JK performed the co-IP. HH performed mass spectrometry and analysed the data. MP, JZ and TM performed immunohistochemistry and analysed the data. AHK, SL and JEL contributed to electrophysiology and analysed the data. FZ, JJC, ACD, BMK and GB provided supervision. JNW and JZ conceived the study, designed the research and provided supervision. All authors prepared the manuscript. JZ and JNW wrote the article with inputs from AHK, HH and MP.

## Conflict of interest

The authors declare that they have no conflict of interest.

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
