## [Review Process File · The EMBO Journal]

Mapping protein interactions of sodium channel Nav1.7 using epitope-tagged gene targeted mice

Alexandros H. Kanellopoulos, Jennifer Koenig, Honglei Huang, Martina Pyrski, Queensta Millet, Stephane Lolignier, Toru Morohashi, Samuel J. Gossage, Maude Jay, John Linley, Georgios Baskozos, Benedikt Kessler, James J. Cox, Annette C. Dolphin, Frank Zufall, John N. Wood and Jing Zhao

Review timeline:

Submission date:	5 April 2017
Editorial Decision:	10 April 2017
Additional Correspondence	13 April 2017
Additional Correspondence	19 April 2017
Resubmission:	24 April 2017
Editorial Decision:	2 June 2017
Revision received:	26 September 2017
Editorial Decision:	7 November 2017
Revision received:	30 November 2017
Accepted:	5 December 2017

Editor: Karin Dumstrei

Transaction Report:

1st Editorial Decision

10 April 2017

Thank you for submitting your manuscript (EMBOJ-2017-96692) to The EMBO Journal. I have now had a chance to read it carefully and to discuss it with my colleagues. I am sorry to say that we cannot offer publication in The EMBO Journal.

Your analysis reports on the identification of Nav1.7 interacting proteins by generating Tap-tagged Nav1.7 KI mice. The KI mice show normal pain behaviour and a detailed analysis is carried out on the expression of the TAP-tagged Nav1.7. TAP-tagged Nav1.7 complexes are identified using mass spec and a number of the candidates are further validated. Further functional data is provided to support that Crmp2 acts as a transporter for Nav1.7.

I appreciate the approach taken to identify Nav1.7 interacting proteins and I see the value of this. However, for consideration here we would need some further follow up analysis on some of the newly identified interacting partners.

Please note that The EMBO Journal only publish a small percentage of the many manuscripts submitted and that we only subject those manuscripts to external review that contribute major conceptual advances.

I thank you for giving us the opportunity to consider this manuscript and I am sorry that we can't be more positive on this occasion.

Additional Correspondence - Authors

13 April 2017

We have been following up our mass spec paper with other experiments. For example we have shown that there is a bona fide interaction between Nav1.7 and mu-opioid receptors mediated by GRIN1 that may be very significant with respect to the potentiation of Opioid receptor signalling in Nav1.7 null mice recently shown. If we add these data would you reconsider our submission?

Additional Correspondence - Editor

13 April 2017

Just want to get back to you before heading out for 4 day Easter break. I am interested in the paper and I like the KI approach very much - but I need some new functional insight or follow up work.

What exactly do you have? Please outline the experiments a bit more. If it is at the same level as the Crmp2 part probably not enough if it is a little bit more then yes.

Additional Correspondence - Authors

19 April 2017

Thanks for your encouraging email. We have added data (Gprin1 - G protein-regulated inducer of neurite outgrowth 1) that physically and functionally link mu-opioid receptors and Nav1.7, that may explain the exciting data of Isensee and Hucho published this year on opioid receptor regulation by Nav1.7. The CRMP2 data is actually entirely novel as this is the first time the physical link between Nav1.7 and crmp2 and the site of action of lacosamide have been unequivocally demonstrated.

Please find attached our revised cover letter and manuscript.

Resubmission

24 April 2017

As we mentioned in the previous letter, we believe this article to be of broad general interest and particularly important for the pain research field.

Importantly, we have already used this mouse to generate new functionally relevant data. For example, we have shown for the first time that Nav1.7 binds to CRMP2, the analgesic target of the drug lacosamide, that down-regulates Nav1.7 current density. Such an interaction has been proposed but this is the first demonstration of this key analgesic related interaction. As Nav1.7 levels have been linked to pain threshold this is very significant. We have also shown a direct interaction between opioid receptors and Nav1.7 mediated by Gprin1. As there is a dramatic upregulation of opioid receptor activity in Nav1.7 KO mice this provides us with a structural link that may help us unravel the mechanisms of opioid receptor control – potentially by sodium.

This paper thus provides an important new resource of protein interactors of sodium channel Nav1.7, as well as new information about the expression pattern of the channel protein together with new functional insights. We believe that this mouse will be taken up by many groups, not only working in the pain field, but also those studying the pancreas and hypothalamus as well as neurotransmission in the CNS. For this reason, we have submitted the paper as a resource tool.

We do hope you will consider this manuscript for EMBO Journal.

2nd Editorial Decision

2 June 2017

Thank you for submitting your manuscript for consideration by the EMBO Journal. It has now been seen by three referees whose comments are shown below.

As you can see from the comments, the referees appreciate that the analysis adds new insight. However, both referees #2 and 3 also find that we need further functional data into some of the

newly identified binding partner. As you know this was also one of my initial concerns with the analysis. So should you be able to significantly extend the functional part and address the other raised concerns then I would like to consider a revised version. It would be good to have a discussion on how to extend the functional analysis and please contact me to discuss this matter further.

I should add that it is EMBO Journal policy to allow only a single major round of revision and that it is therefore important to sort out the raised concerns at this stage.

When preparing your letter of response to the referees' comments, please bear in mind that this will form part of the Review Process File, and will therefore be available online to the community. For more details on our Transparent Editorial Process, please visit our website: http://emboj.embopress.org/about#Transparent_Process

Thank you for the opportunity to consider your work for publication. I look forward to your revision.

REFeree REPORTS

Referee #1:

In this study, Kanellopoulos, Wood, Zhao and colleagues have elegantly mapped protein interaction for one of the most important ion channels in the neurobiology of pain, Nav1.7, using epitope-tagged gene targeted mice. Nav1.7, which underlies both congenital insensitivity to pain and congenital pain syndromes, has become the hottest topic in drug development for chronic intractable pain. And, while several biotech and large pharma are already in clinical trial with quite selective Nav1.7 inhibitors, potential critical side-effects suggest that a better understanding of how Nav1.7 signaling in the nociceptor might suggest novel approaches to pain management. The present study provides an elegant, detailed map of protein interactions for Nav1.7 and how these additional interacting proteins might provide novel targets for the management of currently intractable medically unresponsive pain, with a more advantageous side-effect profile. This study also provides a key data base for others working in pain and sodium channel mechanisms. While well designed and executed, some minor concerns that if addressed might improve the impact of their work are suggested:

In a previous study, Minett et al. (2015), and Isensee et al. (2017) this group of investigators provide evidence that endogenous opioids contribute to insensitivity to pain in humans and mice lacking Nav1.7 (Nat. Comm. 6, 8967 and Sci. Signal., 10 eaah4874). In the present study they have discovered a molecular link that may help explain this potentially important relationship, namely that Nav1.7 regulates opioid receptor efficacy and interacts with G-protein regulated inducer of neurite outgrowth (Gpr11), a mu-opioid receptor binding protein, demonstrating a physical and functional link between Nav1.7 and opioid signaling. I was, however, a bit confused by the discussion of this important set of observations, on page 15. They should provide references for their statement that "opioid receptors are well known to be regulated by sodium," and also for the observation that "Nav1.7 deletion potentiated opioid action substantially."

Does not lacosamide act both at Nav1.7 and Crmp2? Is Crmp2 the, or a target of lacosamide? Has lacosamide been used as an analgesic?

Page 2: "pain behavior" → pain behaviors

Page 2: "synapototagmin-2" → synaptotagmin

If Kanellopoulos and Zhao "contributed equally to directing this research," should they not both be corresponding authors?

Referee #2:

The manuscript by Kanellopoulos et al. is an interesting work, which used an epitope-tagged gene targeted mouse to identify the macromolecular complex associated with the Nav1.7 ion channel. This is particularly relevant regarding the crucial role of this channel in pain highlighted by impressive genetic data in human. The approach is smart, using tandem-affinity purification (TAP), which offers two consecutive affinity purifications to lower background and less contamination (but precisions must be given regarding the methods). Identification of a set of new Nav1.7 associated proteins certainly bears a great potential, both to further explore the already known functions associated with Nav1.7 and to identify new ones.

Overall, the manuscript is well-written and easy to follow, and presents quality data in a visually clear manner. The experiments are well-designed and the discussion is appropriate. However, the manuscript clearly needs to be reinforced by additional data and several points have to be clarified:

> The part of the manuscript dealing with the characterization of a few new interactors of Nav1.7 needs to be reinforced. It is not always clear why some of the interactors have been selected from the MS list (e.g., Scn3b, Syt2, Lat1, Tmed10). However, and as discussed by the authors, these interactions raise interesting questions, like for Syt2 and other synaptic proteins for a possible role of Nav1.7 in neurotransmitter release, Lat1 (the transporter of gabapentin), Tmed10 for Nav1.7 trafficking, or Gprn1 for the functional link between Nav1.7 and opioid receptors. The functional significance and the detailed underlying mechanisms of these interactions are clearly behind the scope of the present work. However, this part remains too descriptive and not strong enough in the present form (with only one experiment of co-immunoprecipitation in vitro in Fig. 5D, plus additional in vitro electrophysiological data for Crmp2 in Fig. 7), and would need either further investigation of some of these interactors, or characterization of new interactors of interest from the MS list.

> The exact experimental approach is confusing for identification of the Nav1.7 complexes. While the manuscript is enmeshing the tandem-affinity purification (TAP) and the KI mice are indeed expressing a TAP-tagged Nav1.7, it is mentioned p 8, line 178 that the complexes that have been analyzed were identified only by single-step affinity purification (ss-AP). The same point was mentioned again in the discussion p12, lines 256-257 and p13, line 284. However, this does not seem to be consistent with the approach described in the Materials and Methods, which indicate p24, line 558 that 'proteins cleaved from Ni-NTA beads after affinity purification...' (i.e., corresponding to TAP purification according to Fig. 5A) have been used for LC-MS/MS. The rationale for using only single-step instead of tandem affinity purification, if it is the case, is not completely clear. The authors mention in the Discussion pages 11 and 12 the advantages of ss-AP versus TAP (especially for transient and dynamic protein-protein interactions). However, it would have been interesting to use both approaches from the same Nav1.7 TAP mice, as least in some tissues, since even if TAP has more experimental constraints compared with ss-AP, it clearly provides improved results with lower background and less contamination.

> Data about the analysis of the Nav1.7 complexes definitely need to be more detailed. It is not clear why only a global analysis is shown in Fig. 6 and Table 1 without any information about potential differences in the different tissues analyzed (or at least in the PNS vs CNS) although TAP-tagged Nav1.7 complexes have been extracted from DRG, spinal cord, olfactory bulb and hypothalamus. Interestingly, the expression pattern of TAP-tagged Nav1.7 showed no FLAG-staining in DRG (contrary to the CNS), which could be due to masking of the tag in the PNS preventing the antibody binding as proposed by the authors. This could be consistent for instance with the presence of different Nav1.7 complexes in the PNS and the CNS (as well as probably in different neuronal populations within the CNS and PNS).

> Regarding the absence of FLAG-staining in DRG of Nav1.7 TAP mice, it could have been interesting to perform control recordings of the Nav1.7 current in primary cultures of DRG neurons from these mice (even if the protein was detected by Western blot with an anti-FLAG antibody).

Minor points:

> Fig. 1, a negative control on HEK293 cells not expressing TAP-tagged Nav1.7 could be shown.

> Fig. 2, legend: In panel B, p 36, line 915, replace "white boxes represent Nav1.7 exons" by "grey

boxes represent Nav1.7 exons", and line 916 "grey box represents TAP tag" by "black box represents TAP tag".

> Fig. 5E: HAT must be replaced by FLAG.

Referee #3:

In the article "Mapping protein interactions of sodium channel Nav1.7 using epitope-tagged gene targeted mice" the authors describe the generation of KI mice harboring a Nav1.7 channel tagged with Tap-tagged epitope (Histidine and Flag tags). Tap-Tagged-Nav1.7 currents in HEK293 cells are identical to wt Nav1.7 currents. KI mice have the same motor ability and identical thermal and mechanical sensitivity than wt litter mates. They used anti Flag antibodies to co-purify nav1.7 partners from pooled tissues from KI and used wt tissues as negative control. LC-MS/MS analysis of the protein complex reveals 1252 proteins present in KI and wt tissues, and the authors claimed that 267 are specific to Nav1.7 complex based on some criteria. Last the authors co expressed one of their candidates, Crmp2, in HEK 293 cell lines stably expressing human Nav1.7, and observed an increase of Nav1.7 current.

Overall the generation and characterization of KI mice are well conducted except some minor points, and are quite convincing concerning the fact that TAP-tagged epitope did not modify Nav1.7 current properties.

Major considerations:

Analyzing protein interacting complexes by immunoprecipitation followed by mass spectrometry generally gives a huge amount of hits, not all specific. The authors did not take advantage of the Tap-Tag epitope that could allow to performed two successive purifications to have "lower background and less contamination" as they say in the discussion paragraph. They performed single step affinity purification (ssAP) and used very loose criteria to short list the proteins susceptible to be part of Nav1.7 interacting complex. Thus some proteins, like Crmp2, are present in all 6 ssAP from wt samples which are supposed to be negative control.

From that list, they validated TAP-tagged Nav1.7 interaction with syt2, lat1 Tmed10 grimp1 Scn2b and Crmp2 in an "in vitro system", with over expression of the candidates in HEK293 cells. Co-IP from endogenous tissues would have been more convincing.

Finally, they further studied Crmp2 in vitro. The presence of Crmp2 strongly enhance Nav1.7 current, an effect blocked by Lacosamide drug. These results not really extend our knowledge of the interplay of Lacosamide, Crmp2 and Voltage gated Sodium channels (VGSC). From these results, the direct interaction between Nav1.7 and Crmp2 claimed in abstract and lines 210; 216; 222, is over stated.

Minor considerations:

*For the behavioral experiment the authors used the same set of mice (7 KI and 7 wt) to perform successively Rotarod, von Frey, Hargreave's and Randal-Sellito tests. Since Hargreave's and Randal-Sellito tests cause painful stimuli the authors should have explained in which order did they have performed the tests, and the delay between two successive test. The ideal situation would have been to run each test on a separate set of mice. They also mixed male and female animals in KI and wt groups. It would have been interesting to test separately males and female.

* Fig 5C why the authors mixed DRG , olfactory bulbs, sciatic nerve, hypothalamus and spinal cord for wt sample. They did not mentioned the proportion of each tissue in the sample. Separated wt tissue should be presented.

*More careful attention should have been brought to the manuscript redaction. For example :

-Fig 5E in the text (line 155-158) and in the figure legend, the anti Flag antibody is cited while HAT is indicated in the figure itself. The staining is not as clearly negative as the authors stated for skin lung and heart; the cerebellum an cortex are clearly negative.

-Line 504 and 505 rephrase for redundant words.

- line 523 the lat1 plasmid reference pEMS1229 # 238146 does not exist in addgene data base. The following plasmids exist: pEMS1229 # 29115, but 3 other plasmids containing the lat1 gen exist in addgene data base.

- Fig.7A is the scale bar correct?

1st Revision - authors' response

26 September 2017

Referee #1:

In this study, Kanellopoulos, Wood, Zhao and colleagues have elegantly mapped protein interaction for one of the most important ion channels in the neurobiology of pain, Nav1.7, using epitope-tagged gene targeted mice. Nav1.7, which underlies both congenital insensitivity to pain and congenital pain syndromes, has become the hottest topic in drug development for chronic intractable pain. And, while several biotech and large pharma are already in clinical trial with quite selective Nav1.7 inhibitors, potential critical side-effects suggest that a better understanding of how Nav1.7 signaling in the nociceptor might suggest novel approaches to pain management. The present study provides an elegant, detailed map of protein interactions for Nav1.7 and how these additional interacting proteins might provide novel targets for the management of currently intractable medically unresponsive pain, with a more advantageous side-effect profile. This study also provides a key data base for others working in pain and sodium channel mechanisms. While well designed and executed, some minor concerns that if addressed might improve the impact of their work are suggested:

In a previous study, Minett et al. (2015), and Isensee et al. (2017) this group of investigators provide evidence that endogenous opioids contribute to insensitivity to pain in humans and mice lacking Nav1.7 (Nat. Comm. 6, 8967 and Sci. Signal., 10 eaah4874). In the present study they have discovered a molecular link that may help explain this potentially important relationship, namely that Nav1.7 regulates opioid receptor efficacy and interacts with G-protein regulated inducer of neurite outgrowth (Gprin1), a mu-opioid receptor binding protein, demonstrating a physical and functional link between Nav1.7 and opioid signaling. I was, however, a bit confused by the discussion of this important set of observations, on page 15. They should provide references for their statement that "opioid receptors are well known to be regulated by sodium," and also for the observation that "Nav1.7 deletion potentiated opioid action substantially."

Response (1): We provided two references on "opioid receptors are well known to be regulated by sodium." (Fenalti, Giguere et al., 2014, Ott, Costa et al., 1988), and one reference on "Nav1.7 deletion potentiated opioid action substantially." (Kanellopoulos, Zhao et al., 2017) in the revised MS.

Does not lacosamide act both at Nav1.7 and Crmp2? Is Crmp2 the, or a target of lacosamide? Has lacosamide been used as an analgesic?

Response (2): Yes, lacosamide (LCM) has dual actions both at Nav1.7 and Crmp2 (Jo & Bean, 2017, Wilson & Khanna, 2015). Our electrophysiological data also shows that 100 uM LCM causes ~24% inhibition of Nav1.7 current density and fully abolishes the up-regulation of Nav1.7 currents by overexpressing Crmp2 (Figure 7). Thus, both Nav1.7 and Crmp2 are the target of LCM.

Antiepileptic drugs have been used for treatment of neuropathic pain. As an antiepileptic drug, LCM has recently been investigated in pain relief and has effects in animal models, although it was not approved for the treatment of painful diabetic peripheral neuropathy by the Food and Drug Administration (Moutal, Chew et al., 2016, Ziegler, Hidvégi et al., 2010).

Page 2: "pain behavior" → pain behaviors

Page 2: "synapototagmin-2" → synaptotagmin

Response (3): We have made these two changes in the revised MS.

If Kanellopoulos and Zhao "contributed equally to directing this research," should they not both be corresponding authors?

Response (4): We deleted this sentence in the revised MS.

Referee #2:

The manuscript by Kanellopoulos et al. is an interesting work, which used an epitope-tagged gene targeted mouse to identify the macromolecular complex associated with the Nav1.7 ion channel. This is particularly relevant regarding the crucial role of this channel in pain highlighted by impressive genetic data in human. The approach is smart, using tandem-affinity purification (TAP), which offers two consecutive affinity purifications to lower background and less contamination (but precisions must be given regarding the methods). Identification of a set of new Nav1.7 associated proteins certainly bears a great potential, both to further explore the already known functions associated with Nav1.7 and to identify new ones.

Overall, the manuscript is well-written and easy to follow, and presents quality data in a visually clear manner. The experiments are well-designed and the discussion is appropriate. However, the manuscript clearly needs to be reinforced by additional data and several points have to be clarified:

> The part of the manuscript dealing with the characterization of a few new interactors of Nav1.7 needs to be reinforced. It is not always clear why some of the interactors have been selected from the MS list (e.g., Scn3b, Syt2, Lat1, Tmed10). However, and as discussed by the authors, these interactions raise interesting questions, like for Syt2 and other synaptic proteins for a possible role of Nav1.7 in neurotransmitter release, Lat1 (the transporter of gabapentin), Tmed10 for Nav1.7 trafficking, or Gprn1 for the functional link between Nav1.7 and opioid receptors. The functional significance and the detailed underlying mechanisms of these interactions are clearly behind the scope of the present work. However, this part remains too descriptive and not strong enough in the present form (with only one experiment of co-immunoprecipitation in vitro in Fig. 5D, plus additional in vitro electrophysiological data for Crmp2 in Fig. 7), and would need either further investigation of some of these interactors, or characterization of new interactors of interest from the MS list.

Response (5): We selected the validated interactors based on 1) function (e.g. neurotransmitter release, trafficking etc), 2) fold change (FC) in MS list (covered whole range, e.g. from FC=1.55 (Syt2) to knockout only (Scn3b)), 3) known (Scn3b and Crmp2) and novel (Syt2, Lat1, Tmed10 and Gprn1) interactors.

We validated these 6 candidates (Scn3b, Crmp2, Syt2, Lat1, Tmed10 and Gprn1) together with 6 more new candidates (AKAP12, Neurofascin, Neurotrimin, Kif5b, Ankyrin G and PEBP1) from the MS list using co-IP with dorsal root ganglia from TAP-tagged Nav1.7 knock-in mice. The results are shown below. We have added the result into Figure 5 as panel G. 3

Figure 5F: The validation of selected Nav1.7 protein-interactor candidates with Nav1.7 endogenous expressing DRG tissue. First, the proteins from DRGs of TAP-tagged Nav1.7 mice were extracted in 1% CHAPS lysis buffer. Nav1.7 complex was then immunoprecipitated by magnetic dynabeads conjugated with anti-FLAG antibody. Twelve Nav1.7 interactor candidates, including Scn3b (32 kDa), Syt2 (44 kDa), Crmp2 (70 kDa), Gprn1 (110 kDa), Lat1 (57 kDa), Tmed10 (21 kDa), Akap12 (191 kDa), Nfasc (138 kDa), Neurotrimin (38 kDa), Kif5b (110 kDa), Ankyrin G (243 kDa) and Pebp1 (23 kDa) were detected with their specific antibodies using Western blotting.

For functional insights, we investigated the function of Nav1.7 in relation to presynaptic neurotransmitter release using immunohistochemistry and co-IP. Existing evidences show that Synaptotagmins (Syt) are involved in presynaptic neurotransmitter release, indicating that Nav1.7 may regulate neurotransmitter release in the peripheral central terminal through Syt2. Therefore, we used immunohistochemistry to detect the distribution of neurotransmitter CGRP and Substance P (SP) in the dorsal horn of the spinal cord in both Nav1.7 knockout mice and littermate wild-type control mice. However, there was no obvious reduction of CGRP or SP in the Lamina I and II in the spinal cord in Nav1.7 knockout mice (see the immunostaining figure below). We think this may be because the synthesis and storage of neurotransmitters may not be affected in the presynapse in Nav1.7 knockout mice. We added this figure as Appendix Figure S1 in the revised MS. 4

Appendix Figure S1. Immunohistochemistry of spinal cords. The cross sections of lumbar spinal cord of Nav1.7 knockout mice (KO) and littermate wild-type control mice (WT) were labelled with anti-CGRP (in red), anti-Substance P (in red) and IB4 (in green). Right panel: left panel merged to middle panel. Scale bar = 250 μ m.

It has been identified that the activity of Cav2.2 involved in neurotransmitter release from presynaptic terminals in pain pathways is regulated by its protein interactor Crmp2 (Brittain et al., 2011, Chi et al., 2009). Our study shows that Crmp2 is also a protein interactor of Nav1.7. Therefore, we tested the possibility that Nav1.7 is involved in neurotransmitter release linked to Cav2.2 through Crmp2 using co-IP in an in vitro system in HEK cells. However, the result showed that Cav2.2 was not immunoprecipitated together with Nav1.7 (see figure below), suggesting Cav2.2 and Nav1.7 may regulate neurotransmitter release independently. We added this result as panel G in Figure 5.

Figure 5G. Co-immunoprecipitation of Nav1.7 with Cav2.2. Left panel shows negative western blot results for pull down of transiently transfected HA-tagged Cav2.2 from Tap-tagged Nav1.7 (HAT antibody for detection) in Tap-tagNav1.7 Hek293 stable cell line. Right panel shows control blot from whole cell lysate of HA-tagged Cav2.2 and Tap-tagged Nav1.7. 5

> *The exact experimental approach is confusing for identification of the Nav1.7 complexes. While the manuscript is enchasing the tandem-affinity purification (TAP) and the KI mice are indeed expressing a TAP-tagged Nav1.7, it is mentioned p 8, line 178 that the complexes that have been analyzed were identified only by single-step affinity purification (ss-AP). The same point was mentioned again in the discussion p12, lines 256-257 and p13, line 284. However, this does not seem to be consistent with the approach described in the Materials and Methods, which indicate p24, line 558 that 'proteins cleaved from Ni-NTA beads after affinity purification...' (i.e., corresponding to TAP purification according to Fig. 5A) have been used for LC-MS/MS.*

The rational for using only single-step instead of tandem affinity purification, if it is the case, is not completely clear. The authors mention in the Discussion pages 11 and 12 the advantages of ss-AP versus TAP (especially for transient and dynamic protein-protein interactions). However, it would have been interesting to use both approaches from the same Nav1.7 TAP mice, as least in some tissues, since even if TAP has more experimental constraints compared with ss-AP, it clearly provides improved results with lower background and less contamination.

Response (6): 'proteins cleaved from Ni-NTA beads after affinity purification...' in line 558 on p24:- this is a mistake. It should be: 'Proteins cleaved from M2 Magnetic FLAG coupled beads after affinity purification'. We have replaced this sentence with the correct one in the revised MS.

Yes, it would have been interesting to use both approaches for the same Nav1.7-TAP mice. However, the data we provide here is useful resource data and we plan to carry out more experiments with different tissues and in different purification conditions as Referee 2 suggested, e.g. ss-AP vs TAP purification in the spinal cord or/and olfactory bulbs in the near future.

> *Data about the analysis of the Nav1.7 complexes definitely need to be more detailed. It is not clear why only a global analysis is shown in Fig. 6 and Table 1 without any information about potential differences in the different tissues analyzed (or at least in the PNS vs CNS) although TAP-tagged Nav1.7 complexes have been extracted from DRG, spinal cord, olfactory bulb and hypothalamus. Interestingly, the expression pattern of TAP-tagged Nav1.7 showed no FLAG-staining in DRG (contrary to the CNS), which could be due to masking of the tag in the PNS preventing the antibody binding as proposed by the authors. This could be consistent for instance with the presence of different Nav1.7 complexes in the PNS and the CNS (as well as probably in different neuronal populations within the CNS and PNS).*

Response (7): Again, the Nav1.7 interactors in different tissues would be very interesting. As an initial step, we focused on using the TAP-tagged Nav1.7 mice combined with ss-AP and MS to map general Nav1.7 interactors as a resource. The next step, we will purify Nav1.7 complexes from different tissues including DRG, spinal cord, olfactory bulbs etc. and then identify the Nav1.7 interactors with MS.

> *Regarding the absence of FLAG-staining in DRG of Nav1.7 TAP mice, it could have been interesting to perform control recordings of the Nav1.7 current in primary cultures of DRG neurons from these mice (even if the protein was detected by Western blot with an anti-FLAG antibody).*

Response (8): Yes, it could have been interesting to perform Nav1.7 current recordings in primary cultured DRG neurons from TAP-tagged Nav1.7 mice. However, technically it is very difficult to do as DRG neurons also express other TTXs channels such as Nav1.1, Nav1.3 and Nav1.6. 6

Minor points:

> Fig. 1, a negative control on HEK293 cells not expressing TAP-tagged Nav1.7 could be shown.

Response (9): We re-did the immunocytochemistry with a negative control on HEK293 cells not expressing TAP-tagged Nav1.7. The result is shown below. We replace Figure 1B with this result.

Figure 1B. Left panel: Representative immunohistochemistry with an anti-FLAG antibody (in Green) on HEK293 cells stably expressing TAP-tagged Nav1.7 (top) and parental HEK293 (bottom). Middle panel: The cell nuclei were stained with DAPI (blue). Right panel: The left panels were merged to the middle panels. Scale bar = 25 μ m.

> Fig. 2, legend: In panel B, p 36, line 915, replace "white boxes represent Nav1.7 exons" by "grey boxes represent Nav1.7 exons", and line 916 "grey box represents TAP tag" by "black box represents TAP tag".

Response (10): We have corrected these mistakes in the revised MS.

> Fig. 5E: HAT must be replaced by FLAG.

Response (11): We have corrected these mistakes in the revised MS.

Referee #3:

In the article "Mapping protein interactions of sodium channel Nav1.7 using epitope-tagged gene targeted mice" the authors describe the generation of KI mice harboring a Nav1.7 channel tagged with Tap-tagged epitope (Histidine and Flag tags). Tap-Tagged-Nav1.7 currents in HEK293 cells are identical to wt Nav1.7 currents. KI mice have the same motor ability and identical thermal and mechanical sensitivity than wt litter mates. They used anti Flag antibodies to co-purify nav1.7 partners from pooled tissues from KI and used wt tissues as negative control. LC-MS/MS analysis of the protein complex reveals 1252 proteins present in KI and wt tissues, and the authors claimed that 267 are specific to Nav1.7 complex based on some criteria. Last the authors co expressed one of their candidates, Crmp2, in HEK 293 cell lines stably expressing human Nav1.7, and observed an increase of Nav1.7 current.

Overall the generation and characterization of KI mice are well conducted except some minor points, and are quite convincing concerning the fact that TAP-tagged epitope did not modify Nav1.7 current properties.

Major considerations:

Analyzing protein interacting complexes by immunoprecipitation followed by mass spectrometry generally gives a huge amount of hits, not all specific. The authors did not take advantage of the Tap-Tag epitope that could allow to performed two successive purifications to have "lower background and less contamination" as they say in the discussion paragraph. They performed single step affinity purification (ssAP) and used very loose criteria to short list the proteins susceptible to be part of Nav1.7 interacting complex. Thus some proteins, like Crmp2, are present in all 6 ssAP from wt samples which are supposed to be negative control.

From that list, they validated TAP-tagged Nav1.7 interaction with syt2, lat1 Tmed10 grimp1 Scn2b and Crmp2 in an "in vitro system", with over expression of the candidates in HEK293 cells. Co-IP from endogenous tissues would have been more convincing.

Finally, they further studied Crmp2 in vitro. The presence of Crmp2 strongly enhance Nav1.7 current, an effect blocked by Lacosamide drug. These results not really extend our knowledge of the interplay of Lacosamide, Crmp2 and Voltage gated Sodium channels (VGSC). From these results, the direct interaction between Nav1.7 and Crmp2 claimed in abstract and lines 210; 216; 222, is over stated.

Response (12):

1) As we mentioned above - Response (6), the data we provided is a general resource of Nav1.7 protein-protein interactors using ss-AP with this newly generated TAP-tagged Nav1.7 knock-in mouse line. We plan to use TAP purification with different tissues in future investigations.

2) We validated 6 candidates (Scn3b, Crmp2, Syt2, Lat1, Tmed10 and Gprn1) that we showed in the previous version of manuscript, together with 6 new candidates (Akap12, Neurofascin, Neurotrimin, Kif5b, Ankyrin G and Pebp1) with co-IP from TAP-tagged Nav1.7 knock-in mice DRG tissue. The result is shown above - Response (5).

3) We modified these sentences as below:

In abstract:

... we demonstrate ~~a direct~~ an interaction between collapsing-response mediator protein (Crmp2) and Nav1.7, showing that the analgesic drug lacosamide regulates Nav1.7 current density.

Line 210:

... We ~~found confirmed~~ that Tap-tagged Nav1.7 binds ~~directly~~ to Crmp2...

Line 216:

... suggesting Crmp2 acts ~~directly~~ as a transporter for Nav1.7...

Line 222:

... Thus we ~~formally demonstrate for the first time a direct~~ an interaction between Nav1.7 and Crmp2

Minor considerations:

**For the behavioral experiment the authors used the same set of mice (7 KI and 7 wt) to perform successively Rotarod, von Frey, Hargreave's and Randal-Sellitto tests. Since Hargreave's and Randal-Sellitto tests cause painful stimuli the authors should have explained in which order did they have performed the tests, and the delay between two successive test. The ideal situation would have been to run each test on a separate set of mice. They also mixed male and female animals in KI and wt groups. It would have been interesting to test separately males and female.*

Response (13): Yes, we used the same set of mice to perform the acute pain behavioural tests. The order we followed was Rotarod, von Frey, Hargreaves and Randall-Selitto tests. We left a one-day gap between the Rotarod, von Frey and Hargreaves' tests, and a 3-day gap between the Hargreaves' test and Randall-Selitto test. It would have been interesting to test the pain behaviours gender-specifically. We will perform them in our further experiments.

** Fig 5C why the authors mixed DRG , olfactory bulbs, sciatic nerve, hypothalamus and spinal cord for wt sample. They did not mentioned the proportion of each tissue in the sample. Separated wt tissue should be presented.*

Response (14): As we mentioned in Response (7), the Nav1.7 interactions in different tissues would have been very interesting. At this stage of the project, we focused on providing a general list of

Nav1.7 interactors as a resource. The next step will be to purify Nav1.7 complexes from different tissues including DRG, spinal cord, olfactory bulb etc. to identify the tissue-specific Nav1.7 interactors. The proportion of each tissue in the sample was all the DRGs, lumbar enlargement of spinal cord, whole olfactory bulbs, both side of sciatic nerve and whole hypothalamus from one mouse. We used pooled wild-type tissues as a negative control to help identify false positive bands.

**More careful attention should have been brought to the manuscript redaction. For example : -Fig 5E in the text (line 155-158) and in the figure legend, the anti Flag antibody is cited while HAT is indicated in the figure itself. The staining is not as clearly negative as the authors stated for skin lung and heart; the cerebellum an cortex are clearly negative.*

Response (15): We corrected the mistake in the revised MS and changed the description of the result as ‘...but not obviously present in cortex, cerebellum, skin, lung, heart and pancreas (Fig 5E).’

-Line 504 and 505 rephrase for redundant words.

Response (16): The redundant words have been removed in the revision:- Centrifugation for 15 min at 14,000 rpm removed tThe nuclear fraction and cell debris were removed by centrifugation at 14,000 rpm for 15 minutes at 4°C.

- line 523 the lat1 plasmid reference pEMS1229 # 238146 does not exist in addgene data base. The following plasmids exist: pEMS1229 # 29115, but 3 other plasmids containing the lat1 gen exist in addgene data base.

Response (17): We put the correct reference in the revised MS.

- Fig.7A is the scale bar correct?

Response (18): We have corrected the scale bar in Fig. 7A.

References:

- Fenalti G, Giguere PM, Katritch V, Huang X-P, Thompson AA, Cherezov V, Roth BL, Stevens RC (2014) Molecular control of δ -opioid receptor signaling. *Nature* 506: 191-9
- Jo S, Bean BP (2017) Lacosamide inhibition of Nav1.7 voltage-gated sodium channels: slow binding to fast-inactivated states. *Molecular Pharmacology*: mol. 116.106401
- Kanellopoulos AH, Zhao J, Emery EC, Wood JN (2017) Intracellular Sodium Regulates Opioid Signalling in Peripheral Sensory Neurons. *bioRxiv*
- Moutal A, Chew LA, Yang X, Wang Y, Yeon SK, Telemi E, Meroueh S, Park KD, Shrinivasan R, Gilbraith KB (2016) (S)-Lacosamide inhibition of CRMP2 phosphorylation reduces postoperative and neuropathic pain behaviors through distinct classes of sensory neurons identified by constellation pharmacology. *Pain*
- Ott S, Costa T, Herz A (1988) Sodium modulates opioid receptors through a membrane component different from G-proteins. Demonstration by target size analysis. *J Biol Chem* 263: 10524-10533
- Wilson SM, Khanna R (2015) Specific binding of lacosamide to collapsin response mediator protein 2 (CRMP2) and direct impairment of its canonical function: implications for the therapeutic potential of lacosamide. *Molecular neurobiology* 51: 599-609
- Ziegler D, Hidvégi T, Gurieva I, Bongardt S, Freynhagen R, Sen D, Sommerville K, Group LSS (2010) Efficacy and safety of lacosamide in painful diabetic neuropathy. *Diabetes Care* 33: 839-841

3rd Editorial Decision

7 November 2017

Thank you for submitting your revised manuscript to The EMBO Journal. I am sorry for the delay in getting back to you with a decision, but I have now received the referee report on your manuscript. As you can see below, the referee appreciates the introduced changes. I am therefore very happy to accept the manuscript for publication here.

 REFEREE REPORT

Referee #2:

The paper has been significantly improved by additional data. Although not absolutely mandatory, the answer to comment #8 (i.e., to perform control recordings of the Nav1.7 current in primary cultures of DRG neurons from Nav1.7 TAP mice) seems however a little short. I agree that DRG neurons also express other TTXs channels but strategies to record Nav1.7 currents in primary cultured DRG neurons have been described, and in addition the use of newly described pharmacological tools (Deuis et al. Sci Rep. 2017, PMID 28106092) could for instance make it rather straightforward.

Anyway, I am sure that this work will provide a valuable contribution to the field and be of great interest to readers of EMBO Journal.

Corresponding Author Name: John N Wood and Jing Zhao

Journal Submitted to: The EMBO Journal

Manuscript Number: EMBOJ-2017-96692R-Q